# Scalar products and norm of Bethe vectors in $\mathfrak{o}_{2n+1}$ invariant integrable models

A. Liashyk[a], S. Pakuliak[b] and E. Ragoucy[b]

[a] *Beijing Institute of Mathematical Sciences and Applications (BIMSA),*
*No. 544, Hefangkou Village Huaibei Town, Huairou District Beijing 101408, China*

[b] *Laboratoire d'Annecy-le-Vieux de Physique Théorique (LAPTh)*
*Chemin de Bellevue, BP 110, F-74941, Annecy-le-Vieux cedex, France*

E-mails: liashyk@bimsa.cn, pakuliak@lapth.cnrs.fr, ragoucy@lapth.cnrs.fr

## Abstract

We compute scalar products of off-shell Bethe vectors in models with $\mathfrak{o}_{2n+1}$ symmetry. The scalar products are expressed as a sum over partitions of the Bethe parameter sets, the building blocks being the so-called highest coefficients. We prove some recurrence relations and a residue theorem for these highest coefficients, and prove that they are consistent with the reduction to $\mathfrak{gl}_n$ invariant models. We also express the norm of on-shell Bethe vectors as a Gaudin determinant.

## 1 Introduction

Integrable systems, characterized by their infinite number of conserved quantities, occupy a central place in mathematical physics. These systems, appearing prominently in models of statistical mechanics and quantum field theory, exhibit remarkable solvability due to their underlying algebraic structures. A key tool for solving these systems is the Bethe Ansatz, first introduced by Hans Bethe in 1931 in the context of the Heisenberg spin chain, which provides exact solutions for eigenstates of certain Hamiltonians [2].

The Bethe Ansatz leads to the construction of Bethe vectors ( [14] for $\mathfrak{gl}_N$ and [21] for $\mathfrak{o}_N$ invariant models), eigenstates expressed in terms of rapidities satisfying the Bethe equations. These vectors are foundational for understanding the spectrum of integrable models and play a central role in calculating physical observables such as correlation functions [3]. The scalar product of Bethe vectors is particularly important, as it provides overlaps between eigenstates, allowing the evaluation of matrix elements of operators. Such calculations are crucial for studying the dynamics and thermodynamics of integrable systems [22].

Depending on the spin chain model one considers, different results have been obtained. For $\mathfrak{gl}_2$ and $\mathfrak{gl}_3$ invariant models, the expression of the scalar product as a sum over partition of the Bethe parameter sets has been obtained in [11] and [20] respectively. The general case of $\mathfrak{gl}_n$ invariant models has been dealt in [7]. The calculation of the norm as a Gaudin determinant has been obtained in [13] for $\mathfrak{gl}_2$, in [20] for $\mathfrak{gl}_3$ and in [8] for $\mathfrak{gl}_{n|m}$. An alternative expression for the norms of eigenstates in $\mathfrak{gl}_n$ based models was calculated using an approach based on the quantized Knizhnik–Zamolodchikov equation [19, 24, 25]. An expression of the scalar product as a determinant, when one of the Bethe vector is on-shell has been computed in [22] for $\mathfrak{gl}_2$ invariant models.

In the present paper, we present the case of $\mathfrak{o}_{2n+1}$ invariant models. We compute the scalar product of Bethe vectors in $\mathfrak{o}_{2n+1}$ invariant models as a sum over partitions. As for other models, the expression makes appear the so-called highest coefficients, and we provide recurrence relations and some residue formula for these coefficients. We also compute the norm of on-shell Bethe vectors as a Gaudin determinant. We show that our results are consistent with the results already obtained for $\mathfrak{gl}_n$ invariant models, and also with the case of $\mathfrak{o}_3$ models, studied in [18].

## 2 Integrable models with $\mathfrak{o}_{2n+1}$ symmetry

**The Yangian $Y(\mathfrak{o}_{2n+1})$.** The models we are considering have a $\mathfrak{o}_{2n+1}$ invariance. They are constructed within the Yangian $Y(\mathfrak{o}_{2n+1})$, defined through a $(2n+1) \times (2n+1)$ monodromy matrix $T(z)$ obeying the celebrated FRT relations [5]:

$$R(u-v)\, T_1(u)\, T_2(v) = T_2(v)\, T_1(u)\, R(u-v)\,, \tag{2.1}$$

where the $R$-matrix for the Yangian $Y(\mathfrak{o}_{2n+1})$ [26] is defined by

$$R(z) = \mathbf{I} \otimes \mathbf{I} + \frac{c}{z}\,\mathbf{P} - \frac{c}{z+c\kappa}\,\mathbf{Q}\,, \qquad \mathbf{P} = \sum_{i,j=-n}^{n} \mathsf{e}_{i,j} \otimes \mathsf{e}_{j,i} \ \text{ and } \ \mathbf{Q} = \sum_{i,j=-n}^{n} \mathsf{e}_{i,j} \otimes \mathsf{e}_{-i,-j} \tag{2.2}$$

with $\kappa = n - 1/2$ and the spectral parameter $z$. In (2.2), we have labeled the indices from $-n$ to $n$, and $\mathsf{e}_{i,j}$ is the elementary matrix with 1 at position $(i,j)$ and 0 elsewhere. We define the transposition as

$$\left(\mathsf{e}_{i,j}\right)^{\mathrm{t}} = \mathsf{e}_{-j,-i}\,. \tag{2.3}$$

Decomposing the monodromy matrix as

$$T(z) = \sum_{i,j=-n}^{n} \mathsf{e}_{i,j} \otimes T_{i,j}(z)\,,$$

we get

$$[T_{i,j}(u), T_{k,l}(v)] = \frac{c}{u-v}\Big(T_{k,j}(v)T_{i,l}(u) - T_{k,j}(u)T_{i,l}(v)\Big)$$
$$+ \frac{c}{u-v+c\kappa} \sum_{p=-n}^{n} \Big(\delta_{k,-i}\, T_{p,j}(u)T_{-p,l}(v) - \delta_{l,-j}\, T_{k,-p}(v)T_{i,p}(u)\Big)\,. \tag{2.4}$$

These relations imply that $T(z)T^t(z+c\kappa)$ is central, and we will impose

$$T(z)^{\mathrm{t}} \cdot T(z+c\kappa) = T(z+c\kappa) \cdot T(z)^{\mathrm{t}} = \mathbf{I}\,. \tag{2.5}$$

Indeed, starting from any model, a simple rescaling of the monodromy matrix by a function $f(z)$, $T(z) \to f(z)T(z)$, will ensure that the condition (2.5) is fulfilled.

**The generalized models.** From the monodromy matrix, one introduces an algebraic transfer matrix

$$\mathcal{T}(z) = \sum_{j=-n}^{n} T_{j,j}(z)\,. \tag{2.6}$$

The FRT relation ensures that $[\mathcal{T}(z), \mathcal{T}(w)] = 0$, so that, upon expansion in the spectral parameter, the transfer matrix generates an Abelian subalgebra of the Yangian.

To get a physical model, one needs to specify a representation of the Yangian algebra. Typically, one takes a tensor of evaluation representations to get a spin chain on a periodic one-dimensional lattice. However, most of the calculations concerning the Bethe vectors can be performed at the algebraic level, leading to the notion of generalized models. In these models, one solely imposes the conditions

$$T_{i,i}(u)|0\rangle = \lambda_i(u)|0\rangle \quad \text{and} \quad T_{i,j}(u)|0\rangle = 0, \ -n \le j < i \le n\,, \tag{2.7}$$

where $\lambda_i(z)$ are arbitrary free functions and $|0\rangle$ is the so-called vacuum state. Due to the relation (2.5), the $\lambda_i(z)$ functions satisfy the relations

$$\lambda_{-j}(z) = \frac{1}{\lambda_j(z_j)} \ \prod_{s=j+1}^{n} \frac{\lambda_s(z_{s-1})}{\lambda_s(z_s)}, \qquad j = 0, 1, \ldots, n\,, \tag{2.8}$$

where $z_s = z - c(s - 1/2)$, $s = 0, 1, \ldots n$.

The 'real' physical models are obtained by giving an explicit form for the functions $\lambda_i(z)$. For example, one can consider

$$\lambda_{-n}(z) = a(z)\left(1 + \frac{c}{z}\right)^L; \quad \lambda_j(z) = a(z)\,, \ -n < j < n; \quad \lambda_n(z) = a(z)\left(1 - \frac{c}{z + c\kappa}\right)^L,$$

$$a(z)\,a(z - c\kappa) = \left(\frac{z^2}{(z+c)(z-c)}\right)^L,$$

where $a(z)$ has been fixed in such a way that (2.5) and (2.8) are satisfied. This choice of $\lambda_j(z)$ corresponds to a spin chain model of $L$ sites, each of them carrying a fundamental $((2n+1)$-dimensional $)$ representation of the $\mathfrak{o}_{2n+1}$ Lie algebra. In this spin chain, the monodromy matrix is realized as a product of $R$ matrices:

$$T_0(z) = a(z)^L\, R_{0L}(z) \cdots R_{01}(z).$$

Let us insist that this is just an example of the models we deal with. In the following, we will keep the functions $\lambda_i(z)$ free, and the monodromy matrix as algebraic, just obeying the relations (2.4), (2.5) and (2.7).

**Notations.** We will use different sets of parameters, indexed by an integer $s$ running from 0 to $n - 1$ and called the color of the set: $\bar{u}^{(s)} = \{u_1^{(s)}, \ldots, u_{r_s}^{(s)}\}$. As a convention, sets (and subsets) will be always noted with a bar, as in the preceeding example. The cardinality of these sets will be noted $|\bar{u}^{(s)}| = r_s$. Collections of such sets will be noted as $\{\bar{u}^{(i)}, \bar{u}^{(i+1)}, ..., \bar{u}^{(j)}\} = \{\bar{u}^{(s)}\}_{s=i}^{j}$ and the full collection of sets as $\bar{u} = \{\bar{u}^{(0)}, \bar{u}^{(1)}, ..., \bar{u}^{(n-1)}\} = \{\bar{u}^{(s)}\}_{s=0}^{n-1}$. As an additional notation, $\bar{v}_k$ will be the subset complementary to the element $v_k$ in the set $\bar{v}$: $\bar{v}_k^{(s)} = \bar{v}^{(s)} \setminus \{v_k^{(s)}\}$.

We will also use sums over partitions of these sets in two or three subsets, e.g. $\bar{u}^s \vdash \{\bar{u}_{\mathrm{I}}^s, \bar{u}_{\mathrm{II}}^s\}$, where $\bar{u}_{\mathrm{I}}^s$ and $\bar{u}_{\mathrm{II}}^s$ are disjoint (possibly empty) subsets such that $\bar{u}_{\mathrm{I}}^s \cup \bar{u}_{\mathrm{II}}^s = \bar{u}^s$. Let us stress that the subsets may be empty, although in general this does not occur in a partition. For example,

$$\sum_{\bar{u}=\{u_1,u_2\}\vdash\{\bar{u}_{\mathrm{I}},\bar{u}_{\mathrm{II}}\}} G(\bar{u}_{\mathrm{I}}, \bar{u}_{\mathrm{II}}) = G(\varnothing, \bar{u}) + G(u_1, u_2) + G(u_2, u_1) + G(\bar{u}, \varnothing).$$

We define the functions

$$f(u,v) = \frac{u-v+c}{u-v} \ , \quad \mathfrak{f}(u,v) = \frac{u-v+c/2}{u-v} \ , \quad g(u,v) = \frac{c}{u-v} \ ,$$

$$h(u,v) = \frac{f(u,v)}{g(u,v)} = \frac{u-v+c}{c} \ , \quad \gamma_s(u,v) = \begin{cases} \mathfrak{f}(u,v) & \text{when } s = 0 \ , \\ \dfrac{g(u,v)}{h(v,u)} & \text{otherwise} \ , \end{cases} \tag{2.9}$$

$$\alpha_s(u) = \frac{\lambda_s(u)}{\lambda_{s+1}(u)} \ .$$

To lighten the presentation of the results, we will use the following convention. For any function depending on one or two variables, if a set appears as a variable, then one has to consider the product of this function at all element of the set. For instance for a set $\bar{u}^{(s)}$ of cardinality $r_s$, $f(\bar{u}^{(s)}, \bar{v}) = \prod_{\ell=1}^{r_s} f(u_\ell^{(s)}, \bar{v})$. As a rule, we will also set $f(\varnothing, \bar{v}) = 1$.

**Bethe vectors.**   Usually, Bethe vectors are defined as eigenvectors of the transfer matrix

$$\mathcal{T}(z) \, \mathbb{B}(\bar{u}) = \tau(z; \bar{u}) \, \mathbb{B}(\bar{u}) \ ,$$

$$\tau(z; \bar{u}) = \lambda_0(z) f(\bar{u}^{(0)}, z_0) f(z, \bar{u}^{(0)}) +$$

$$+ \sum_{s=1}^n \lambda_s(z) f(\bar{u}^{(s)}, z) f(z, \bar{u}^{(s-1)}) + \lambda_{-s}(z) f(\bar{u}^{(s-1)}, z_{s-1}) f(z_s, \bar{u}^{(s)})$$

provided the Bethe equations are obeyed

$$\alpha_0(u_k^{(0)}) = \frac{\mathfrak{f}(u_k^{(0)}, \bar{u}_k^{(0)}) \, f(\bar{u}^{(1)}, u_k^{(0)})}{\mathfrak{f}(\bar{u}_k^{(0)}, u_k^{(0)})} \ , \qquad k = 1, ..., r_0 \ ,$$

$$\alpha_s(u_k^{(s)}) = \frac{f(u_k^{(s)}, \bar{u}_k^{(s)}) \, f(\bar{u}^{(s+1)}, u_k^{(s)})}{f(\bar{u}_k^{(s)}, u_k^{(s)}) \, f(u_k^{(s)}, \bar{u}^{(s-1)})} \ , \quad k = 1, ..., r_s \ , \ s = 1, ..., n-1. \tag{2.10}$$

We will call these vectors *on-shell Bethe vectors* because their Bethe parameters obey the Bethe equations (2.10).

If we do not require the Bethe equations to be satisfied, we will call the corresponding vector an *off-shell Bethe vector*. Bethe vectors (on-shell or off-shell) are polynomials of the monodromy matrix entries acting on the vacuum vector state:

$$\mathbb{B}(\bar{u}) = \mathcal{P}\big(\{T_{i,j}(u_k^{(s)}), \ -n \le i < j \le n, \ k = 1, \dots, r_s, \ 0 \le s \le n-1\}\big) \, |0\rangle \equiv \mathcal{B}(\bar{u}) \, |0\rangle.$$

$\mathcal{B}(\bar{u})$ is called a *pre-Bethe vector*.

Besides the transposition (2.3) we define a usual matrix transposition

$$\big(\mathsf{e}_{i,j}\big)' = \mathsf{e}_{j,i} \ . \tag{2.11}$$

We use the same notation for the transposition anti-morphism $(\cdot)'$ in the algebra of monodromy matrix entries determined by this matrix transposition

$$\big(T(u)\big)' = \sum_{i,j=-n}^n \mathsf{e}_{i,j} \otimes \big(T_{i,j}(u)\big)' = \sum_{i,j=-n}^n \big(\mathsf{e}_{i,j}\big)' \otimes T_{i,j}(u) = \sum_{i,j=-n}^n \mathsf{e}_{i,j} \otimes T_{j,i}(u) \ . \tag{2.12}$$

Being extended to the vacuum vectors $\langle 0| = |0\rangle'$ and $|0\rangle = \langle 0|'$ this transposition anti-morphism allows to define the dual Bethe vectors as follows

$$\mathbb{C}(\bar{u}^{(0)}, \dots, \bar{u}^{(n-1)}) = \big(\mathbb{B}(\bar{u}^{(0)}, \dots, \bar{u}^{(n-1)})\big)' \ , \tag{2.13}$$

where according to (2.12) $\big(T_{i,j}(u)\big)' = T_{j,i}(u)$. All the definitions given above also apply to dual Bethe vectors.

# 3 Properties of Bethe vectors

In this section, we present different properties of off-shell Bethe vectors (BVs), that will be needed for the calculation of their scalar products.

## 3.1 Action formula on BVs

To describe the action of the monodromy entry $T_{i,j}(z)$ on the off-shell Bethe vector $\mathbb{B}(\bar{u})$ we define the extended sets $\bar{w}^{(s)} = \bar{u}^{(s)} \cup \{z, z_s\}$ for $s = 0, 1, \ldots, n-1$ with $z_s = z - c(s-1/2)$. The action formula has been given in equation (3.5) of [15] as a sum over partitions $\bar{w}^{(s)} \vdash \{\bar{w}_{\text{I}}^{(s)}, \bar{w}_{\text{II}}^{(s)}, \bar{w}_{\text{III}}^{(s)}\}$, with cardinalities

$$\left|\bar{w}_{\text{I}}^{(s)}\right| = \begin{cases} 2, & s < i \le n, \\ 1, & -s \le i \le s, \\ 0, & -n \le i < -s, \end{cases} \quad \text{and} \quad \left|\bar{w}_{\text{III}}^{(s)}\right| = \begin{cases} 0, & s < j \le n, \\ 1, & -s \le j \le s, \\ 2, & -n \le j < -s, \end{cases} \tag{3.1a}$$

$$\left|\bar{w}_{\text{I}}^{(s)}\right| + \left|\bar{w}_{\text{II}}^{(s)}\right| + \left|\bar{w}_{\text{III}}^{(s)}\right| = \left|\bar{w}^{(s)}\right|. \tag{3.1b}$$

**Remark 3.1** *The condition* (3.1b) *should be automatically satisfied since we have a partition. However, in some particular cases (depending on $i$, $j$ and the cardinalities of $\bar{u}$), the conditions* (3.1a) *may contradict this condition: in that case, the corresponding term should be discarded, leading to a vanishing of the action formula for the corresponding $T_{i,j}(z)$. An example of such case is given below, for the reduction to the $\mathfrak{gl}_n$ case.*

For a partition of the sets $\bar{w}^{(s)}$ into pairs of subsets $\bar{w}_{\text{I}}^{(s)}$ and $\bar{w}_{\text{II}}^{(s)}$ we define the function

$$\Omega(\bar{w}_{\text{I}}|\bar{w}_{\text{II}}) = \prod_{s=0}^{n-1} \gamma_s(\bar{w}_{\text{I}}^{(s)}, \bar{w}_{\text{II}}^{(s)}) \, \frac{h(\bar{w}_{\text{II}}^{(s+1)}, \bar{w}_{\text{I}}^{(s)})}{g(\bar{w}_{\text{I}}^{(s+1)}, \bar{w}_{\text{II}}^{(s)})} \, . \tag{3.2}$$

We also define the functions $\Phi_{i,j}(\bar{w})$

$$\Phi_{i,j}(\bar{w}) = - \, \sigma_i \sigma_{-j} \, \frac{g(z_1, \bar{u}^{(0)})}{\kappa \, h(z, \bar{u}^{(0)})} \, \Omega(\bar{w}_{\text{I}}|\bar{w}_{\text{II}}) \, \Omega(\bar{w}_{\text{II}}|\bar{w}_{\text{III}}) \, \Omega(\bar{w}_{\text{I}}|\bar{w}_{\text{III}}) \, , \tag{3.3}$$

where

$$\sigma_i = \begin{cases} -1, & \text{if } i < 1, \\ 1, & \text{if } i \ge 1. \end{cases} \tag{3.4}$$

Let us stress that $\Phi_{ij}(\bar{w})$ does not depend on the functions $\lambda_s$. Note also that although the expression of $\Phi_{ij}(\bar{w})$ does not seem to depend on $i$ and $j$, this dependence is hidden in the cardinalities of the partitions, as detailed in (3.1). The function $\Phi_{i,j}(\bar{w})$ depends on an additional boundary set $\bar{w}^{(n)} = \{z, z_n\}$ with a fixed partition

$$\bar{w}_{\text{I}}^{(n)} = \{z_n\} \, , \qquad \bar{w}_{\text{II}}^{(n)} = \varnothing \, , \qquad \bar{w}_{\text{III}}^{(n)} = \{z\} \, . \tag{3.5}$$

The action of the monodromy matrix entries on BVs reads [15]

$$T_{i,j}(z) \cdot \mathbb{B}(\bar{u}) = \lambda_n(z) \sum_{\text{part}} \left( \prod_{s=0}^{n-1} \alpha_s(\bar{w}_{\text{III}}^{(s)}) \right) \Phi_{i,j}(\bar{w}) \, \mathbb{B}(\bar{w}_{\text{II}}) \, , \tag{3.6}$$

where the sum goes over partition $\bar{w}^{(s)} \vdash \{\bar{w}_{\mathrm{I}}^{(s)}, \bar{w}_{\mathrm{II}}^{(s)}, \bar{w}_{\mathrm{III}}^{(s)}\}$ with the subsets cardinalities given by (3.1).

Relying on the Yangian $Y(\mathfrak{o}_{2n+1})$, we assume that the monodromy matrix has the following dependence on the formal parameter $u$

$$T_{i,j}(z) = \delta_{i,j} + \sum_{m \geq 0} T_{i,j}[m] \, (z/c)^{-m-1} \tag{3.7}$$

and we define the zero modes $\mathsf{T}_{i,j} \equiv T_{i,j}[0]$ as

$$\mathsf{T}_{i,j} = \lim_{z \to \infty} \frac{z}{c} \left( T_{i,j}(z) - \delta_{i,j} \right). \tag{3.8}$$

These zero modes satisfy the commutation relations of the $\mathfrak{o}_{2n+1}$ algebra

$$[\mathsf{T}_{i,j}, \mathsf{T}_{k,l}] = \left( \delta_{i,l} \, \mathsf{T}_{k,j} - \delta_{j,k} \, \mathsf{T}_{i,l} \right) - \left( \delta_{j,-l} \, \mathsf{T}_{k,-i} - \delta_{i,-k} \, \mathsf{T}_{-j,l} \right). \tag{3.9}$$

Let $\mathsf{t}_s$ for $s = 0, 1, \ldots, n-1$ be the operators

$$\mathsf{t}_s = \sum_{i=s+1}^{n} \left( T_{i,i}[0] - \lambda_i[0] \right) \tag{3.10}$$

such that $\mathsf{t}_s \, |0\rangle = 0$ and $\langle 0| \, \mathsf{t}_s = 0$. The adjoint action of the operators $\mathsf{t}_s$ on the monodromy entry $T_{i,j}(z)$ reads

$$[\mathsf{t}_s, T_{i,j}(z)] = \sum_{\ell=s+1}^{n} \left( \delta_{\ell,j} - \delta_{\ell,i} + \delta_{\ell,-i} - \delta_{\ell,-j} \right) T_{i,j}(z). \tag{3.11}$$

For $i < j$, relation (3.11) always yields a non-negative eigenvalue equal to either 0, 1 or 2 depending on relations between the indices $s$, $i$, and $j$.

**Proposition 3.1** *The operators $\mathsf{t}_s$ being applied to the BVs and dual BVs measure the cardinalities of their Bethe parameters:*

$$\mathsf{t}_s \cdot \mathbb{B}(\bar{u}) = r_s \, \mathbb{B}(\bar{u}), \qquad \mathbb{C}(\bar{u}) \cdot \mathsf{t}_s = r_s \, \mathbb{C}(\bar{u}), \quad s = 0, \ldots, n-1. \tag{3.12}$$

*Proof:* This proposition can be proved by induction on the total cardinality $|\bar{u}|$, using (3.11) and the recurrence relations for the Bethe vectors (3.14) described in the next section. We will prove only the first equation in (3.12) since the second one results from the application of the anti-morphism (2.13) to the first. The base of the induction is the case when all sets are empty: according to the definition (3.10) $\mathsf{t}_s \cdot \mathbb{B}(\varnothing) = \mathsf{t}_s \cdot |0\rangle = 0$ for $s = 0, 1, \ldots, n-1$. Assume now that the first equation in (3.12) is valid for $|\bar{u}| = r$ and any $s$, and apply an operator $\mathsf{t}_s$ to both sides of the recurrence relation (3.14). Using (3.11) and the description of the cardinalities (3.15) we find that

$$\mathsf{t}_s \cdot \mathbb{B}(\bar{u}, z^{(\ell)}) = (r_s + \delta_{s,\ell}) \, \mathbb{B}(\bar{u}, z^{(\ell)}).$$

Since $|\{\bar{u}, z^{(\ell)}\}| = r + 1$, this proves the induction and finishes the proof of the proposition 3.1.    $\square$

## 3.2 Recurrence relations for BVs

To lighten the presentation we introduce notation

$$\left(\bar{u}, z^{(\ell)}\right) = \left(\left\{\bar{u}^{(s)}\right\}_{s=0}^{\ell-1}, \left\{\bar{u}^{(\ell)}, z\right\}, \left\{\bar{u}^{(s)}\right\}_{s=\ell+1}^{n-1}\right), \tag{3.13}$$

for any fixed $\ell = 0, 1, \ldots, n - 1$. It can be shown that the BVs in the $\mathfrak{o}_{2n+1}$ invariant models obey several types of recurrence relations [17]. For a given $\ell$ the recurrence relations relevant for this paper take the general form[1]

$$\mathbb{B}(\bar{u}, z^{(\ell)}) = \sum_{i=-n}^{\ell} \sum_{j=\ell+1}^{n} \sum_{\text{part}} \left(\prod_{s=\ell+1}^{j-1} \alpha_s(\bar{u}_{\mathrm{III}}^{(s)})\right) \Psi_{i,j}^{(\ell)}(\bar{u}, z) \frac{T_{i,j}(z) \cdot \mathbb{B}(\bar{u}_{\mathrm{II}})}{\lambda_{\ell+1}(z)}, \tag{3.14}$$

where the sum is on partition of $\bar{u}^{(s)} \vdash \{\bar{u}_{\mathrm{I}}^{(s)}, \bar{u}_{\mathrm{II}}^{(s)}, \bar{u}_{\mathrm{III}}^{(s)}\}$, with cardinalities depending on $i$ and $j$ as follows

$$\text{for} \quad 0 \leq s < \ell: \qquad |\bar{u}_{\mathrm{I}}^{(s)}| = \begin{cases} 2, & i < -s \leq 0, \\ 1, & -s \leq i \leq s, \\ 0, & s < i, \end{cases} \qquad |\bar{u}_{\mathrm{III}}^{(s)}| = 0,$$

$$\text{for} \quad s = \ell: \qquad |\bar{u}_{\mathrm{I}}^{(\ell)}| = \begin{cases} 1, & i < -\ell, \\ 0, & -\ell \leq i, \end{cases} \qquad |\bar{u}_{\mathrm{III}}^{(\ell)}| = 0, \tag{3.15}$$

$$\text{for} \quad \ell < s \leq n-1: \qquad |\bar{u}_{\mathrm{I}}^{(s)}| = \begin{cases} 1, & i < -s, \\ 0, & -s \leq i, \end{cases} \qquad |\bar{u}_{\mathrm{III}}^{(s)}| = \begin{cases} 1, & s < j, \\ 0, & j \leq s. \end{cases}$$

The function $\Psi_{i,j}^{(\ell)}(\bar{u}, z)$ is a rational function that does not depend on the functions $\alpha_s$. Its explicit expression depends on the indices $i$, $j$ and $\ell$ as follows.

**In the case $\ell > 0$,** they take the form

$$\Psi_{i,j}^{(\ell)}(\bar{u}, z) = \sigma_{i+1} \frac{g(z, \bar{u}_{\mathrm{I}}^{(\ell-1)}) h(\bar{u}_{\mathrm{I}}^{(\ell)}, z) g(\bar{u}_{\mathrm{III}}^{(\ell+1)}, z)}{g(z, \bar{u}^{(\ell-1)}) h(z, \bar{u}^{(\ell)}) h(\bar{u}^{(\ell)}, z) g(\bar{u}^{(\ell+1)}, z)} \Omega(\bar{u}_{\mathrm{I}} | \bar{u}_{\mathrm{II}}) \Omega(\bar{u}_{\mathrm{I,II}} | \bar{u}_{\mathrm{III}}), \tag{3.16}$$

where notation $\bar{u}_{\mathrm{I,II}}$ means the union $\bar{u}_{\mathrm{I}} \cup \bar{u}_{\mathrm{II}}$. According to our convention for the products of the rational functions $\Omega(\bar{u}_{\mathrm{I,II}}, \bar{u}_{\mathrm{III}}) = \Omega(\bar{u}_{\mathrm{I}} | \bar{u}_{\mathrm{III}}) \Omega(\bar{u}_{\mathrm{II}} | \bar{u}_{\mathrm{III}})$. Note that for $\ell > 0$, we must have $n \geq 2$.

**In the case $\ell = 0$,** we have

$$\Psi_{i,j}^{(0)}(\bar{u}, z) = \sigma_{i+1} \frac{g(z_0, \bar{u}_{\mathrm{I}}^{(0)}) g(\bar{u}_{\mathrm{III}}^{(1)}, z)}{g(z_0, \bar{u}^{(0)}) h(z, \bar{u}^{(0)}) g(\bar{u}^{(1)}, z)} \Omega(\bar{u}_{\mathrm{I}} | \bar{u}_{\mathrm{II}}) \Omega(\bar{u}_{\mathrm{I,II}} | \bar{u}_{\mathrm{III}}) \tag{3.17}$$

with the cardinalities

$$|\bar{u}_{\mathrm{I}}^{(0)}| = \begin{cases} 1, & \text{if } i < 0, \\ 0, & \text{otherwise}, \end{cases} \qquad |\bar{u}_{\mathrm{III}}^{(0)}| = 0,$$

$$|\bar{u}_{\mathrm{I}}^{(s)}| = \begin{cases} 1, & \text{if } s < -i, \\ 0, & \text{otherwise}, \end{cases} \qquad |\bar{u}_{\mathrm{III}}^{(s)}| = \begin{cases} 1, & \text{if } s < j, \\ 0, & \text{otherwise}, \end{cases} \qquad s = 1, \ldots, n-1. \tag{3.18}$$

The recurrence relation (3.14) with the function (3.17) is valid also in the case $n = 1$ and coincides with the recurrence relation given in [18].

---

[1]In [17] another type recurrence relations was also considered when the set $\bar{u}^{(\ell)}$ is extended by the shifted parameter $z_\ell$. In this paper we will not use these so-called shifted recurrence relations.

## 3.3 Recursion formula for dual BVs

From the recurrence relations (3.14), applying the transposition anti-morphism (2.11), we get recurrence relations for dual BVs. For a given $\ell = 0, 1, ..., n-1$, they take the general form

$$\mathbb{C}(\bar{v}, z^{(\ell)}) = \sum_{j=-n}^{\ell} \sum_{i=\ell+1}^{n} \sum_{\text{part}} \Big( \prod_{s=\ell+1}^{i-1} \alpha_s(\bar{v}_{\text{III}}^{(s)}) \Big) \, \Psi_{j,i}^{(\ell)}(\bar{v}, z) \, \frac{\mathbb{C}(\bar{v}_{\text{II}}) \cdot T_{i,j}(z)}{\lambda_{\ell+1}(z)} \,, \qquad (3.19)$$

with cardinalities

$$
\begin{aligned}
\text{for} \quad 0 \le s < \ell : \qquad & |\bar{v}_{\text{I}}^{(s)}| = \begin{cases} 2, & j < -s \le 0, \\ 1, & -s \le j \le s, \\ 0, & s < j, \end{cases} \qquad |\bar{v}_{\text{III}}^{(s)}| = 0, \\[2mm]
\text{for} \quad s = \ell : \qquad & |\bar{v}_{\text{I}}^{(\ell)}| = \begin{cases} 1, & j < -\ell, \\ 0, & -\ell \le j, \end{cases} \qquad\quad |\bar{v}_{\text{III}}^{(\ell)}| = 0, \\[2mm]
\text{for} \quad \ell < s \le n-1 : \qquad & |\bar{v}_{\text{I}}^{(s)}| = \begin{cases} 1, & j < -s, \\ 0, & -s \le j, \end{cases} \qquad |\bar{v}_{\text{III}}^{(s)}| = \begin{cases} 1, & s < i, \\ 0, & i \le s. \end{cases}
\end{aligned}
\qquad (3.20)
$$

The functions $\Psi_{j,i}^{(\ell)}(\bar{v}, z)$ are deduced from the expressions (3.16) and (3.17).

## 3.4 Coproduct property for BVs

We remind the well-known expression for the standard coproduct for the monodromy matrix:

$$\Delta\Big(T_{i,j}(u)\Big) = \sum_{k} T_{k,j}(u) \otimes T_{i,k}(u) = \sum_{k} T_{k,j}^{[1]}(u) \, T_{i,k}^{[2]}(u) \,.$$

Since pre-Bethe vectors $\mathcal{B}(\bar{u})$ are certain polynomials of the monodromy matrix entries $T_{i,j}(u_k^{(s)})$ one can address the question to calculate their coproduct. In general, this is a quite non-trivial combinatorial problem which was solved for $\mathfrak{gl}_2$ invariant BVs in the 80's [13] and lately for $\mathfrak{gl}_n$ invariant BVs in [24]. In the latter paper a trace formula for the pre-Bethe vectors was used to prove the coproduct properties of $\mathfrak{gl}_n$ invariant BVs. Lately on, an alternative approach to obtain the presentation for the $U_q(\mathfrak{gl}_n)$ invariant pre-Bethe vectors in terms of the Cartan-Weyl generators of the quantum affine algebra was proposed [12]. Both approaches, the trace formula and the Cartan-Weyl presentation, produce slightly different expressions for the pre-Bethe vectors. Nevertheless these expressions are different only by terms which are annihilated on the vacuum state, but the calculation of the coproduct of BVs in the Cartan-Weyl approach is rather simple. It uses the relation between the standard and Drinfeld coproducts of the simple root Cartan-Weyl generators in quantum affine algebra proved in [4].

An extension of the Cartan-Weyl approach for the description of the off-shell Bethe vectors in $\mathfrak{gl}(m|n)$ invariant models was achieved in [6] and for $\mathfrak{o}_{2n+1}$ invariant models in [15]. In order to develop this description one has to replace the Yangian by its double. In the current presentation for the Yangian double $DY(\mathfrak{o}_{2n+1})$, the Bethe vectors take the form of a projection of currents:

$$\mathbb{B}(\bar{u}) = \mathcal{B}(\bar{u})|0\rangle = \mathcal{P}^+\Big(\mathbb{F}(\bar{u})\Big)|0\rangle \qquad (3.21)$$

where $\mathcal{P}^+$ is a projection on the intersection of different Borel subalgebras in the Yangian double and $\mathbb{F}(\bar{u})$ is a product of Cartan-Weyl generators in a certain order. For more

details on the projection method in $\mathfrak{o}_{2n+1}$ models, see e.g. [15]. In this framework, the coproduct of pre-Bethe vectors can be computed assuming that there is the same relation between standard and Drinfeld coproduct for the Yangian doubles as for the quantum affine algebras and using the Drinfeld coproduct of the currents. One gets

$$\Delta\Big(\mathcal{B}(\bar{u})\Big) \sim \sum_{\text{part}} \Omega(\bar{u}_{\text{II}}|\bar{u}_{\text{I}})\,\mathcal{B}(\bar{u}_{\text{I}}) \otimes \mathcal{B}(\bar{u}_{\text{II}}) \prod_{s=0}^{n-1} \Big(\mathbb{I} \otimes T_{ss}(z)(\bar{u}_{\text{I}}^{(s)})\,T_{s+1,s+1}(z)(\bar{u}_{\text{I}}^{(s)})^{-1}\Big),\ \ (3.22)$$

where the equivalence $\sim$ means an equality modulo terms which are annihilated on the tensor product of the vacuum states $|0\rangle \otimes |0\rangle$. The sum runs over partitions $\bar{u}^{(s)} \vdash \{\bar{u}_{\text{I}}^{(s)}, \bar{u}_{\text{II}}^{(s)}\}$. Since, as far as we know, the trace formula for $\mathfrak{o}_{2n+1}$ invariant Bethe vector is not known, the Cartan-Weyl generators approach is a unique way to find formulas for the pre-Bethe vectors and their coproducts in $\mathfrak{o}_{2n+1}$ invariant integrable models.

In the following we will use the formula (3.22) in a composite model, where the monodromy matrix is splitted into two submodel monodromy matrices: $\Delta(T(u)) = T^{[2]}(u)\,T^{[1]}(u)$. Correspondingly the eigenvalues $\lambda_s(u)$ factorize in the same way, and for instance $\alpha_s(u) = \alpha_s^{[1]}(u)\alpha_s^{[2]}(u)$. The coproduct property allows to compute the Bethe vector of the composite model as the product of Bethe vectors corresponding to the submodels:

$$\Delta\Big(\mathbb{B}(\bar{u})\Big) = \sum_{\text{part}} \Omega(\bar{u}_{\text{II}}|\bar{u}_{\text{I}})\,\mathbb{B}^{[1]}(\bar{u}_{\text{I}})\,\mathbb{B}^{[2]}(\bar{u}_{\text{II}}) \prod_{s=0}^{n-1} \alpha_s^{[2]}(\bar{u}_{\text{I}}^{(s)}),\ \ \ \ \ \ (3.23)$$

where $\mathbb{B}^{[1]}(\bar{u}_{\text{I}})$ and $\mathbb{B}^{[2]}(\bar{u}_{\text{II}})$ are BVs for the submodels based on $T^{[1]}(u)$ and $T^{[2]}(u)$ respectively. Note that the Bethe equations (2.10) can be rewritten in the form

$$\prod_{s=0}^{n-1} \alpha_s(\bar{u}_{\text{I}}^{(s)}) = \frac{\Omega(\bar{u}_{\text{I}}|\bar{u}_{\text{II}})}{\Omega(\bar{u}_{\text{II}}|\bar{u}_{\text{I}})}, \ \ \ \text{for any partitions}\ \ \ \bar{u}^{(s)} \vdash \{\bar{u}_{\text{I}}^{(s)}, \bar{u}_{\text{II}}^{(s)}\}, \ \ \ s = 0,\ldots,n-1.$$
$$(3.24)$$

This implies that for on-shell BVs, there is an alternative form to the coproduct formula:

$$\Delta\Big(\mathbb{B}(\bar{u})\Big) = \sum_{\text{part}} \frac{\Omega(\bar{u}_{\text{I}}|\bar{u}_{\text{II}})\,\mathbb{B}^{[1]}(\bar{u}_{\text{I}})\,\mathbb{B}^{[2]}(\bar{u}_{\text{II}})}{\prod_{s=0}^{n-1} \alpha_s^{[1]}(\bar{u}_{\text{I}}^{(s)})} = \frac{\Delta^{op}\Big(\mathbb{B}(\bar{u})\Big)}{\prod_{s=0}^{n-1} \alpha_s^{[1]}(\bar{u}^{(s)})},\ \ \ \ \ \ (3.25)$$

where $\Delta^{\text{op}} = \sigma \circ \Delta$ with $\sigma(A^{[1]}\,B^{[2]}) = A^{[2]}\,B^{[1]}$.

# 4  Sum formula for the scalar products in $Y(\mathfrak{o}_{2n+1})$ models

The sum formula for the scalar product of BVs in $Y(\mathfrak{o}_{2n+1})$ models expresses the scalar product as a sum over partition of rational functions and product of $\alpha_s$ functions. It has been proven in the case of $Y(\mathfrak{gl}_2)$ models in [10], and then generalized in [20] for $Y(\mathfrak{gl}_3)$ and in [7,9] for $Y(\mathfrak{gl}_{m|n})$ and $U_q(\widehat{\mathfrak{gl}}_n)$ models. The goal of this section is to prove it for $Y(\mathfrak{o}_{2n+1})$ models and is summarized in the theorem 4.6.

We consider the scalar product $S(\bar{v}|\bar{u}) = \mathbb{C}(\bar{v})\,\mathbb{B}(\bar{u})$ of BVs associated to $\mathfrak{o}_{2n+1}$ models.

**Lemma 4.1**   *(i) The scalar product $S(\bar{v}|\bar{u})$ is symmetric in the exchange $\bar{u} \leftrightarrow \bar{v}$.*

*(ii) It is also invariant under any permutation within the set $\bar{u}^{(s)}$ and within the set $\bar{v}^{(s)}$.*

*(iii) $S(\bar{v}|\bar{u}) = 0$ whenever there is at least one color $s$ such that $|\bar{v}^{(s)}| \neq |\bar{u}^{(s)}|$.*

*Proof:* Since $S(\bar{v}|\bar{u})$ is a scalar, we have

$$S(\bar{v}|\bar{u}) = S(\bar{v}|\bar{u})' = \mathbb{B}(\bar{u})' \, \mathbb{C}(\bar{v})' = \mathbb{C}(\bar{u}) \, \mathbb{B}(\bar{v}) = S(\bar{u}|\bar{v}).$$

Hence the first claim. The second one is a direct consequence of the same property for BVs themselves. Finally, using operators $\mathsf{t}_s$ defined by (3.10) from

$$\mathbb{C}(\bar{v}) \left( \mathsf{t}_s \, \mathbb{B}(\bar{u}) \right) = |\bar{u}^{(s)}| \, \mathbb{C}(\bar{v}) \, \mathbb{B}(\bar{u}) = \left( \mathbb{C}(\bar{v}) \, \mathsf{t}_s \right) \mathbb{B}(\bar{u}) = |\bar{v}^{(s)}| \, \mathbb{C}(\bar{v}) \, \mathbb{B}(\bar{u}) \, ,$$

we get the last claim. $\qquad\qquad\qquad\qquad\qquad\qquad\qquad\qquad\qquad\qquad\square$

**Lemma 4.2** *The scalar product $S(\bar{v}|\bar{u})$ depends on the functions $\alpha_s(u_j^{(s)})$ and $\alpha_s(v_k^{(s)})$, $1 \leq j,k \leq r_s$, $s = 0,\ldots,n-1$, but not on the functions $\alpha_s(u_j^{(p)})$ or $\alpha_s(v_k^{(p)})$, $p \neq s$. Moreover, each Bethe parameter $u_j^{(s)}$ or $v_k^{(s)}$ occurs at most once in the function $\alpha_s$.*

The proof of this lemma can be found in appendix A. $\qquad\qquad\qquad\qquad\qquad\square$

**Proposition 4.3** *The scalar product $S(\bar{v}|\bar{u})$ can be written as*

$$S(\bar{v}|\bar{u}) = \sum_{part} W(\bar{v}_{\mathrm{I}}, \bar{v}_{\mathrm{II}}|\bar{u}_{\mathrm{I}}, \bar{u}_{\mathrm{II}}) \prod_{s=0}^{n-1} \alpha_s(\bar{v}_{\mathrm{I}}^{(s)}) \, \alpha_s(\bar{u}_{\mathrm{II}}^{(s)}) \, , \qquad (4.1)$$

*where the sum is taken over partitions $\bar{u}^{(s)} \vdash \{\bar{u}_{\mathrm{I}}^{(s)}, \bar{u}_{\mathrm{II}}^{(s)}\}$ and $\bar{v}^{(s)} \vdash \{\bar{v}_{\mathrm{I}}^{(s)}, \bar{v}_{\mathrm{II}}^{(s)}\}$ with $|\bar{v}_{\mathrm{I}}^{(s)}| = |\bar{u}_{\mathrm{I}}^{(s)}|$, $s = 0,\ldots,n-1$.*

*The functions $W(\bar{v}_{\mathrm{I}}, \bar{v}_{\mathrm{II}}|\bar{u}_{\mathrm{I}}, \bar{u}_{\mathrm{II}})$ do not depend on the eigenvalues $\lambda_j$, $j = 0,\ldots n$. As such, they do not depend on the model under consideration, they depend only on the R-matrix.*

*Proof:* From lemma 4.2, we know that the scalar product is a sum of terms with functions $\alpha_s$ involving at most once each Bethe parameter $u_j^{(s)}$ or $v_k^{(s)}$. This can be realized as a sum over partitions as written in the expression (4.1) with coefficients independent from the $\lambda$'s. Then, it remains to prove that the partitions obey the equalities $|\bar{v}_{\mathrm{I}}^{(s)}| = |\bar{u}_{\mathrm{I}}^{(s)}|$. For such a purpose, we consider a composite model corresponding to a splitting $T(u) = T^{[2]}(u) \, T^{[1]}(u)$ of the monodromy matrix, with $\alpha_s(u) = \alpha_s^{[1]}(u)\alpha_s^{[2]}(u)$.

From the coproduct property for BVs (3.23), using

$$(\cdot)' \otimes (\cdot)' \circ \Delta = \Delta^{\mathrm{op}} \circ (\cdot)' \, ,$$

where $(\cdot)'$ is the transposition anti-morphism defined by (2.12), we deduce

$$\Delta(\mathbb{C}(\bar{v})) = \sum_{part} \Omega(\bar{v}_{\mathrm{I}}|\bar{v}_{\mathrm{II}}) \, \mathbb{C}^{[1]}(\bar{v}_{\mathrm{I}}) \, \mathbb{C}^{[2]}(\bar{v}_{\mathrm{II}}) \prod_{s=0}^{n-1} \alpha_s^{[1]}(\bar{v}_{\mathrm{II}}^{(s)}) \, , \qquad (4.2)$$

where $\Omega(\bar{v}_{\mathrm{I}}|\bar{v}_{\mathrm{II}})$ is defined as in (3.2). The sum in (4.2) runs over partitions $\bar{v}^{(s)} \vdash \{\bar{v}_{\mathrm{I}}^{(s)}, \bar{v}_{\mathrm{II}}^{(s)}\}$ and $\mathbb{C}^{[1]}(\bar{v}_{\mathrm{I}})$ and $\mathbb{C}^{[2]}(\bar{v}_{\mathrm{II}})$ are dual BVs for the submodels based on $T^{[1]}(v)$ and $T^{[2]}(v)$ respectively. Then, the scalar product $\mathbb{C}(\bar{v}) \, \mathbb{B}(\bar{u})$ takes the form

$$S(\bar{v}|\bar{u}) = \Delta(S(\bar{v}|\bar{u})) = \sum_{part} \Omega(\bar{u}_{\mathrm{II}}|\bar{u}_{\mathrm{I}}) \, \Omega(\bar{v}_{\mathrm{I}}|\bar{v}_{\mathrm{II}}) \, S^{[1]}(\bar{v}_{\mathrm{I}}|\bar{u}_{\mathrm{I}}) \, S^{[2]}(\bar{v}_{\mathrm{II}}|\bar{u}_{\mathrm{II}}) \prod_{s=0}^{n-1} \alpha_s^{[2]}(\bar{u}_{\mathrm{I}}^{(s)}) \, \alpha_s^{[1]}(\bar{v}_{\mathrm{II}}^{(s)}) \, .$$
$$(4.3)$$

Since, from lemma 4.1, for the scalar products $S^{[1]}$ and $S^{[2]}$ to be non-zero, we need to have $|\bar{v}_{\mathrm{I}}^{(s)}| = |\bar{u}_{\mathrm{I}}^{(s)}|$ and $|\bar{v}_{\mathrm{II}}^{(s)}| = |\bar{u}_{\mathrm{II}}^{(s)}|$, we get the result. $\qquad\square$

**Proposition 4.4** *The coefficients $W(\bar{v}_{\mathrm{I}}, \bar{v}_{\mathrm{II}}|\bar{u}_{\mathrm{I}}, \bar{u}_{\mathrm{II}})$ in $S(\bar{v}|\bar{u})$ can be expressed as*

$$W(\bar{v}_{\mathrm{I}}, \bar{v}_{\mathrm{II}}|\bar{u}_{\mathrm{I}}, \bar{u}_{\mathrm{II}}) = \ \Omega(\bar{u}_{\mathrm{I}}|\bar{u}_{\mathrm{II}})\,\Omega(\bar{v}_{\mathrm{II}}|\bar{v}_{\mathrm{I}})\ Z(\bar{v}_{\mathrm{I}}|\bar{u}_{\mathrm{I}})\,\overline{Z}(\bar{v}_{\mathrm{II}}|\bar{u}_{\mathrm{II}})\,, \tag{4.4}$$

*where $\Omega$ is defined in* (3.2), *and the highest coefficients $Z(\bar{v}_{\mathrm{I}}|\bar{u}_{\mathrm{I}})$ and $\overline{Z}(\bar{v}_{\mathrm{II}}|\bar{u}_{\mathrm{II}})$ are defined by*

$$Z(\bar{v}_{\mathrm{I}}|\bar{u}_{\mathrm{I}}) = W(\bar{v}_{\mathrm{I}}, \varnothing|\bar{u}_{\mathrm{I}}, \varnothing) \quad and \quad \overline{Z}(\bar{v}_{\mathrm{II}}|\bar{u}_{\mathrm{II}}) = W(\varnothing, \bar{v}_{\mathrm{II}}|\varnothing, \bar{u}_{\mathrm{II}})\,,$$

*with the normalisation $W(\varnothing, \varnothing|\varnothing, \varnothing) = 1$.*

*Proof:* To prove this property, we use the fact that the coefficients $W(\bar{v}_{\mathrm{I}}, \bar{v}_{\mathrm{II}}|\bar{u}_{\mathrm{I}}, \bar{u}_{\mathrm{II}})$ do not depend on the model under consideration. We fix two partitions $\bar{u}^{(s)} \vdash \{\bar{u}_{\mathrm{i}}^{(s)}, \bar{u}_{\mathrm{ii}}^{(s)}\}$ and $\bar{v}^{(s)} \vdash \{\bar{v}_{\mathrm{i}}^{(s)}, \bar{v}_{\mathrm{ii}}^{(s)}\}$ such that $|\bar{u}_{\mathrm{i}}^{(s)}| = |\bar{v}_{\mathrm{i}}^{(s)}|$ and $|\bar{u}_{\mathrm{ii}}^{(s)}| = |\bar{v}_{\mathrm{ii}}^{(s)}|$, and consider a model[2] where we have

$$\alpha_s^{[1]}(z) = 0 \quad \text{if} \quad z \in \bar{v}_{\mathrm{ii}}^{(s)} \quad \text{and} \quad \alpha_s^{[2]}(z) = 0 \quad \text{if} \quad z \in \bar{u}_{\mathrm{i}}^{(s)}\,. \tag{4.5}$$

Considering expression (4.1), since $\alpha_s(z) = 0$ when $z \in \bar{v}_{\mathrm{ii}}^{(s)} \cup \bar{u}_{\mathrm{i}}^{(s)}$, we obtain

$$S(\bar{v}|\bar{u}) = W(\bar{v}_{\mathrm{i}}, \bar{v}_{\mathrm{ii}}|\bar{u}_{\mathrm{i}}, \bar{u}_{\mathrm{ii}}) \prod_{s=0}^{n-1} \alpha_s(\bar{v}_{\mathrm{i}}^{(s)})\,\alpha_s(\bar{u}_{\mathrm{ii}}^{(s)})\,. \tag{4.6}$$

On the other hand, we see that in (4.3), for the product $\prod_{s=0}^{n-1} \alpha_s^{[2]}(\bar{u}_{\mathrm{I}}^{(s)})\,\alpha_s^{[1]}(\bar{v}_{\mathrm{II}}^{(s)})$ to be non-zero, we need to have $\bar{u}_{\mathrm{I}}^{(s)} \subset \bar{u}_{\mathrm{ii}}^{(s)}$ and $\bar{v}_{\mathrm{II}}^{(s)} \subset \bar{v}_{\mathrm{i}}^{(s)}$. But because of the constraint on the cardinalities of the subsets $|\bar{u}_{\mathrm{I}}^{(s)}| = |\bar{v}_{\mathrm{I}}^{(s)}|$ and $|\bar{u}_{\mathrm{i}}^{(s)}| = |\bar{v}_{\mathrm{i}}^{(s)}|$, it leads to $\bar{u}_{\mathrm{I}}^{(s)} = \bar{u}_{\mathrm{ii}}^{(s)}$ and $\bar{v}_{\mathrm{II}}^{(s)} = \bar{v}_{\mathrm{i}}^{(s)}$, so that

$$S(\bar{v}|\bar{u}) = S^{[1]}(\bar{v}_{\mathrm{ii}}|\bar{u}_{\mathrm{ii}})\,S^{[2]}(\bar{v}_{\mathrm{i}}|\bar{u}_{\mathrm{i}})\,\Omega(\bar{u}_{\mathrm{i}}|\bar{u}_{\mathrm{ii}})\Omega(\bar{v}_{\mathrm{ii}}|\bar{v}_{\mathrm{i}}) \prod_{s=0}^{n-1} \alpha_s^{[2]}(\bar{u}_{\mathrm{ii}}^{(s)})\,\alpha_s^{[1]}(\bar{v}_{\mathrm{i}}^{(s)})\,. \tag{4.7}$$

Now, we can use again relation (4.1) to compute the scalar products $S^{[1]}(\bar{v}_{\mathrm{ii}}|\bar{u}_{\mathrm{ii}})$ and $S^{[2]}(\bar{v}_{\mathrm{i}}|\bar{u}_{\mathrm{i}})$. We detail the calculation of $S^{[1]}(\bar{v}_{\mathrm{ii}}|\bar{u}_{\mathrm{ii}})$, the other case being similar. We need to perform partitions $\bar{u}_{\mathrm{ii}}^{(s)} \vdash \{\bar{u}_{\mathrm{iii}}^{(s)}, \bar{u}_{\mathrm{iv}}^{(s)}\}$ and $\bar{v}_{\mathrm{ii}}^{(s)} \vdash \{\bar{v}_{\mathrm{iii}}^{(s)}, \bar{v}_{\mathrm{iv}}^{(s)}\}$, which will make appear terms with a factor $\prod_{s=0}^{n-1} \alpha_s^{[1]}(\bar{v}_{\mathrm{iii}}^{(s)})\,\alpha_s^{[1]}(\bar{u}_{\mathrm{iv}}^{(s)})$. However, because of (4.5), only the partition corresponding to $\bar{v}_{\mathrm{iii}}^{(s)} = \varnothing$ will have a non-zero contribution. Since $|\bar{v}_{\mathrm{iii}}^{(s)}| = |\bar{u}_{\mathrm{iii}}^{(s)}|$, we get

$$S^{[1]}(\bar{v}_{\mathrm{ii}}|\bar{u}_{\mathrm{ii}}) = W(\varnothing, \bar{v}_{\mathrm{ii}}|\varnothing, \bar{u}_{\mathrm{ii}}) \prod_{s=0}^{n-1} \alpha_s^{[1]}(\bar{u}_{\mathrm{ii}}^{(s)}) \quad \text{and} \quad S^{[2]}(\bar{v}_{\mathrm{i}}|\bar{u}_{\mathrm{i}}) = W(\bar{v}_{\mathrm{i}}, \varnothing|\bar{u}_{\mathrm{i}}, \varnothing) \prod_{s=0}^{n-1} \alpha_s^{[2]}(\bar{v}_{\mathrm{i}}^{(s)})\,. \tag{4.8}$$

Plugging these expressions in (4.7) and comparing it with (4.6) gives relation (4.4). $\quad\square$

**Corollary 4.5** *The highest coefficient $Z(\bar{v}|\bar{u})$ is invariant under any permutation of $\bar{v}^{(s)}$ and any permutation of $\bar{u}^{(s)}$, for any $s$.*

*It obey the symmetry relation*

$$Z(\bar{v}|\bar{u}) = \overline{Z}(\bar{u}|\bar{v})\,. \tag{4.9}$$

*This relation is sufficient to ensure the symmetry of the scalar product given in* (i) *of lemma 4.1.*

---

[2]The existence of such model is shown in appendix B.

*Proof:* From relation (4.8), we deduce that there is a model where for instance

$$S^{[2]}(\bar{v}_{\rm i}|\bar{u}_{\rm i}) = Z(\bar{v}_{\rm i}|\bar{u}_{\rm i}) \prod_{s=0}^{n-1} \alpha_s^{[2]}(\bar{v}_{\rm i}^{(s)}).$$

The scalar product being invariant under the permutations, we deduce that in this model the highest coefficient is also invariant. Since the highest coefficient is independent from the choice of the model, we conclude it is always invariant.

Starting from symmetry relation $S(\bar{u}|\bar{v}) = S(\bar{v}|\bar{u})$ and using the sum formula, we get

$$\sum_{\rm part} W(\bar{v}_{\rm I}, \bar{v}_{\rm II}|\bar{u}_{\rm I}, \bar{u}_{\rm II}) \prod_{s=0}^{n-1} \alpha_s(\bar{v}_{\rm I}^{(s)})\, \alpha_s(\bar{u}_{\rm II}^{(s)}) = \sum_{\rm part} W(\bar{u}_{\rm I}, \bar{u}_{\rm II}|\bar{v}_{\rm I}, \bar{v}_{\rm II}) \prod_{s=0}^{n-1} \alpha_s(\bar{u}_{\rm I}^{(s)})\, \alpha_s(\bar{v}_{\rm II}^{(s)})\,.$$

Since $\alpha$'s are free functionals, we can project this relation on $\prod_{s=0}^{n-1} \alpha_s(\bar{v}_{\rm I}^{(s)})\, \alpha_s(\bar{u}_{\rm II}^{(s)})$ for a fixed but arbitrary partition. This leads to

$$W(\bar{v}_{\rm I}, \bar{v}_{\rm II}|\bar{u}_{\rm I}, \bar{u}_{\rm II}) = W(\bar{u}_{\rm II}, \bar{u}_{\rm I}|\bar{v}_{\rm II}, \bar{v}_{\rm I}), \tag{4.10}$$

which is valid for any partition. Setting $\bar{v}_{\rm II} = \bar{u}_{\rm II} = \varnothing$ in this relation, we get (4.9). Note that from proposition 4.4, one deduces that relation (4.9) is sufficient to ensure (4.10) and implies the symmetry of the scalar product. $\qquad\square$

Gathering the results obtained in this section we get as a final formula for the scalar product

**Theorem 4.6** *The scalar product $S(\bar{v}|\bar{u}) = \mathbb{C}(\bar{v})\,\mathbb{B}(\bar{u})$ obeys the following sum formula*

$$S(\bar{v}|\bar{u}) = \sum_{part} \Omega(\bar{u}_{\rm I}|\bar{u}_{\rm II})\,\Omega(\bar{v}_{\rm II}|\bar{v}_{\rm I})\; Z(\bar{v}_{\rm I}|\bar{u}_{\rm I})\, Z(\bar{u}_{\rm II}|\bar{v}_{\rm II}) \prod_{s=0}^{n-1} \alpha_s(\bar{v}_{\rm I}^{(s)})\, \alpha_s(\bar{u}_{\rm II}^{(s)})\,, \tag{4.11}$$

*where the sum is taken over partitions $\bar{u}^{(s)} \vdash \{\bar{u}_{\rm I}^{(s)}, \bar{u}_{\rm II}^{(s)}\}$ and $\bar{v}^{(s)} \vdash \{\bar{v}_{\rm I}^{(s)}, \bar{v}_{\rm II}^{(s)}\}$ with $|\bar{v}_{\rm I}^{(s)}| = |\bar{u}_{\rm I}^{(s)}|$, $s = 0, \ldots, n-1$. $Z(\bar{v}|\bar{u})$ is called the highest coefficient and does not depend on the choice of the model, i.e. it does not depend on the functions $\alpha_s$.*

Remark that the relation (4.11) looks formally identical to the sum formula for $Y(\mathfrak{gl}_n)$ models, with however the restriction that the terms associated to $s = 0$ have a different form, see definitions (2.9). This implies that when $\bar{u}^{(0)} = \bar{v}^{(0)} = \varnothing$, we get exactly the sum formula for the $Y(\mathfrak{gl}_n)$ models, as expected.

# 5 Recursion relations

From the recurrence relations detailed in section 3.2, we can deduce some recurrence relations obeyed by the scalar product. Their iterative action allows to express scalar products of $Y(\mathfrak{o}_{2n+1})$ models in terms of scalar products of $Y(\mathfrak{gl}_{n-\ell-1}) \otimes Y(\mathfrak{o}_{2\ell+3})$ models. To lighten the presentation, we will use the notation $(\bar{v}, z^{(\ell)})$ introduced in (3.13).

**Proposition 5.1** *For $\ell \geq 0$, the scalar product obeys the following recurrence relation, with $|\bar{u}^{(\ell)}| = |\bar{v}^{(\ell)}| + 1$ and $|\bar{u}^{(s)}| = |\bar{v}^{(s)}|$ otherwise.*

$$S(\bar{v}, z^{(\ell)}|\bar{u}) = \sum_{j=-n}^{\ell} \sum_{i=\ell+1}^{n} \sum_{\rm part} \frac{\left(\prod_{s=\ell+1}^{i-1} \alpha_s(\bar{v}_{\rm III}^{(s)})\right) \left(\prod_{s=0}^{n-1} \alpha_s(\bar{w}_{\rm III}^{(s)})\right)}{\prod_{s=\ell+1}^{n-1} \alpha_s(z)} \tag{5.1}$$

$$\times\, \Phi_{i,j}(\bar{w})\, \Psi_{j,i}^{(\ell)}(\bar{v}, z) S(\bar{v}_{\rm II}|\bar{w}_{\rm II})\,,$$

*where the functions $\Phi_{i,j}(\bar{w})$ and $\Psi_{j,i}^{(\ell)}(\bar{v}, z)$ are given in (3.3), (3.16) and (3.17). The sum is on partitions $\bar{w}^{(s)} = \{\bar{u}^{(s)}, z, z_s\} \vdash \{\bar{w}_{\mathrm{I}}^{(s)}, \bar{w}_{\mathrm{II}}^{(s)}, \bar{w}_{\mathrm{III}}^{(s)}\}$ and $\bar{v}^{(s)} \vdash \{\bar{v}_{\mathrm{I}}^{(s)}, \bar{v}_{\mathrm{II}}^{(s)}, \bar{v}_{\mathrm{III}}^{(s)}\}$ with cardinalities given in (3.1) and (3.20).*

*Proof:* Direct consequence of the recurrence relation (3.19) and the action formula (3.6). We used the relation

$$\frac{\lambda_{\ell+1}(z)}{\lambda_n(z)} = \prod_{s=\ell+1}^{n-1} \alpha_s(z).$$

$\square$

Note that because of the scalar product $S(\bar{v}_{\mathrm{II}}|\bar{w}_{\mathrm{II}})$ in the rhs of the recurrence relations, we must also have $|\bar{v}_{\mathrm{II}}| = |\bar{w}_{\mathrm{II}}|$, which could lead to additional constraints on the partitions. We show it is not the case when $\ell = n-1$, the proof for the other values of $\ell$ being similar. To compare the two sets $\bar{v}_{\mathrm{II}}$ and $\bar{w}_{\mathrm{II}}$, we use

$$|\bar{w}_{\mathrm{II}}^{(s)}| = \begin{cases} |\bar{w}^{(s)}| - 4 = |\bar{u}^{(s)}| - 2, & j < -s \leq 0, \\ |\bar{w}^{(s)}| - 3 = |\bar{u}^{(s)}| - 1, & |j| \leq s < n, \\ |\bar{w}^{(s)}| - 2 = |\bar{u}^{(s)}|, & s < j, \\ 0, & s = n, \end{cases} \tag{5.2}$$

that is deduced from the cardinalities (3.1a). Moreover, the lemma 4.1-($iii$) implies that

$$|\bar{v}^{(s)}| = |\bar{u}^{(s)}|, \quad s \leq n-2 \quad \text{and} \quad |\bar{v}^{(n-1)}| = |\bar{u}^{(n-1)}| - 1. \tag{5.3}$$

It allows to rewrite (3.1a) as

$$\text{For} \quad s \leq n-2: \qquad |\bar{w}_{\mathrm{II}}^{(s)}| = \begin{cases} |\bar{v}^{(s)}| - 2, & s < -j, \\ |\bar{v}^{(s)}| - 1, & |j| \leq s, \\ |\bar{v}^{(s)}|, & s < j. \end{cases}$$

$$\text{For} \quad s = n-1: \qquad |\bar{w}_{\mathrm{II}}^{(n-1)}| = \begin{cases} |\bar{v}^{(n-1)}| - 1, & j = -n, \\ |\bar{v}^{(n-1)}|, & |j| \leq n-1. \end{cases} \tag{5.4}$$

On the other hand, from (3.20), we deduce that

$$\text{For} \quad s \leq n-2: \qquad |\bar{v}_{\mathrm{II}}^{(s)}| = \begin{cases} |\bar{v}^{(s)}| - 2, & j < -s \leq 0, \\ |\bar{v}^{(s)}| - 1, & |j| \leq s, \\ |\bar{v}^{(s)}|, & s < j. \end{cases}$$

$$\text{For} \quad s = n-1: \qquad |\bar{v}_{\mathrm{II}}^{(n-1)}| = \begin{cases} |\bar{v}^{(n-1)}|, & |j| \leq n-1, \\ |\bar{v}^{(n-1)}| - 1, & j = n, \end{cases} \tag{5.5}$$

which shows that indeed we have $|\bar{v}_{\mathrm{II}}| = |\bar{w}_{\mathrm{II}}|$. Then, the only constraints on partitions in (5.1) are given by (3.1) and (3.20).

## 5.1 Recursion for the highest coefficient

There are different recursion relations for highest coefficients, depending on the color on which we perform the recursion. It also depends on the way we compute the recursion, which amounts to say whether we deal with $Z(\bar{v}|\bar{u})$ or $\overline{Z}(\bar{v}|\bar{u}) = Z(\bar{u}|\bar{v})$.

**Proposition 5.2** *We remind the notation (3.13). The highest coefficient obeys the recurrence relation*

$$Z(\bar{u}|\bar{v}, z^{(\ell)}) = \sum_{j=-n}^{\ell} \sum_{\text{part}} \Phi_{\ell+1,j}(\bar{x}) \, \Psi_{j,\ell+1}^{(\ell)}(\bar{v}, z) \, Z(\bar{x}_{\mathrm{II}}|\bar{v}_{\mathrm{II}}) \,, \tag{5.6}$$

*where*

$$\bar{x}^{(s)} = \begin{cases} \bar{u}^{(s)} \,, & 0 \le s \le \ell, \\ \{\bar{u}^{(s)}, z\} \,, & \ell + 1 \le s \le n - 1, \end{cases} \tag{5.7}$$

*and the sum is on partitions* $\bar{x}^{(s)} \vdash \{\bar{x}_{\mathrm{II}}^{(s)}, \bar{x}_{\mathrm{III}}^{(s)}\}$ *and* $\bar{v}^{(s)} \vdash \{\bar{v}_{\mathrm{I}}^{(s)}, \bar{v}_{\mathrm{II}}^{(s)}\}$ *with cardinalities*

$$\text{For} \quad 0 \le s < \ell: \qquad |\bar{v}_{\mathrm{I}}^{(s)}| = \begin{cases} 2 \,, & j < -s \,, \\ 1 \,, & -s \le j \le s \,, \\ 0 \,, & s < j \,, \end{cases} \qquad |\bar{x}_{\mathrm{III}}^{(s)}| = \begin{cases} 2 \,, & j < -s \,, \\ 1 \,, & -s \le j \le s \,, \\ 0 \,, & s < j \,, \end{cases}$$

$$\text{For} \quad \ell \le s \le n - 1: \quad |\bar{v}_{\mathrm{I}}^{(s)}| = \begin{cases} 1 \,, & j < -s \,, \\ 0 \,, & -s \le j \,, \end{cases} \qquad |\bar{x}_{\mathrm{III}}^{(s)}| = \begin{cases} 2 \,, & j < -s \,, \\ 1 \,, & -s \le j \le s \,, \end{cases} \tag{5.8}$$

*with the convention* $\bar{v}^{(n)} = \varnothing$ *and* $\bar{x}^{(n)} = \bar{x}_{\mathrm{III}}^{(n)} = \{z\}$. *The function* $\Phi_{\ell+1,j}(\bar{x})$ *is given by[3]*

$$\Phi_{\ell+1,j}(\bar{x}) = g(z, \bar{u}^{(\ell)}) \, h(\bar{u}^{(\ell+1)}, z) \, \Omega(\bar{x}_{\mathrm{II}}|\bar{x}_{\mathrm{III}}) \,, \tag{5.9}$$

*while the function* $\Psi_{j,\ell+1}^{(\ell)}(\bar{v}, z)$ *reads for* $\ell > 0$

$$\Psi_{j,\ell+1}^{(\ell)}(\bar{v}, z) = \frac{\Omega(\bar{v}_{\mathrm{I}}|\bar{v}_{\mathrm{II}})}{g(z, \bar{v}_{\mathrm{II}}^{(\ell-1)}) \, h(z, \bar{v}^{(\ell)}) \, h(\bar{v}_{\mathrm{II}}^{(\ell)}, z) \, g(\bar{v}^{(\ell+1)}, z)} \,, \tag{5.10}$$

*and for* $\ell = 0$

$$\Psi_{j,1}^{(0)}(\bar{v}, z) = \frac{g(z_0, \bar{v}_{\mathrm{I}}^{(0)}) \, \Omega(\bar{v}_{\mathrm{I}}|\bar{v}_{\mathrm{II}})}{\mathfrak{f}(z_0, \bar{v}^{(0)}) \, g(\bar{v}^{(1)}, z)} \,. \tag{5.11}$$

*Proof:* We consider expressions modulo terms that contain at least a $\alpha_s(v_k^{(s)})$ or a $\alpha_s(z)$ or a $\alpha_s(z_s)$ term. We note $\simeq$ this equivalence relation. The sum formula (4.11) leads to

$$S(\bar{v}, z^{(\ell)}|\bar{u}) \simeq Z(\bar{u}|\bar{v}, z^{(\ell)}) \prod_{s=0}^{n-1} \alpha_s(\bar{u}^{(s)}) \,. \tag{5.12}$$

Starting from the recurrence relation (5.1) and using (4.11) for the scalar product $S(\bar{v}_{\mathrm{II}}|\bar{w}_{\mathrm{II}})$, the scalar product $S(\bar{v}, z^{(\ell)}|\bar{u})$ also reads

$$S(\bar{v}, z^{(\ell)}|\bar{u}) = \sum_{j=-n}^{\ell} \sum_{i=\ell+1}^{n} \sum_{\text{part}} \frac{\left(\prod_{s=\ell+1}^{i-1} \alpha_s(\bar{v}_{\mathrm{III}}^{(s)})\right) \left(\prod_{s=0}^{n-1} \alpha_s(\bar{w}_{\mathrm{III}}^{(s)})\right)}{\prod_{s=\ell+1}^{n-1} \alpha_s(z)} \Phi_{i,j}(\bar{w}) \, \Psi_{j,i}^{(\ell)}(\bar{v}, z) S(\bar{v}_{\mathrm{II}}|\bar{w}_{\mathrm{II}})$$

$$\simeq \sum_{j=-n}^{\ell} \sum_{i=\ell+1}^{n} \sum_{\text{part}} \frac{\left(\prod_{s=\ell+1}^{i-1} \alpha_s(\bar{v}_{\mathrm{III}}^{(s)})\right) \left(\prod_{s=0}^{n-1} \alpha_s(\bar{w}_{\mathrm{III}}^{(s)})\right)}{\prod_{s=\ell+1}^{n-1} \alpha_s(z)} \Phi_{i,j}(\bar{w}) \, \Psi_{j,i}^{(\ell)}(\bar{v}, z) \prod_{s=0}^{n-1} \alpha_s(\bar{w}_{\mathrm{II}}^{(s)}) \, Z(\bar{w}_{\mathrm{II}}|\bar{v}_{\mathrm{II}}) \,.$$

---

[3]Strictly speaking, there should be a factor $-\sigma_{-j}$ in $\Phi_{\ell+1,j}(\bar{x})$ and a factor $\sigma_{j+1}$ in $\Psi_{j,\ell+1}^{(\ell)}(\bar{v}, z)$. However, in the product $\Phi_{\ell+1,j}(\bar{x}) \, \Psi_{j,\ell+1}^{(\ell)}(\bar{v}, z)$ these terms cancel since $-\sigma_{-j}\sigma_{j+1} = 1$, so that we discarded them.

To compare the two expressions, we need to select the partitions such that

$$|\bar{v}_{\mathrm{III}}^{(s)}| = 0 \quad \text{for} \quad \ell + 1 \le s \le i - 1 \,, \tag{5.13a}$$

$$\{\bar{w}_{\mathrm{II}}^{(s)}, \bar{w}_{\mathrm{III}}^{(s)}\} = \bar{u}^{(s)} \quad \text{for} \quad 0 \le s \le \ell \,, \tag{5.13b}$$

$$\{\bar{w}_{\mathrm{II}}^{(s)}, \bar{w}_{\mathrm{III}}^{(s)}\} = \{\bar{u}^{(s)}, z\} \quad \text{for} \quad \ell + 1 \le s \le n - 1 \,. \tag{5.13c}$$

Looking at the cardinalities (3.20) for $\bar{v}$, one sees that $|\bar{v}_{\mathrm{III}}^{(s)}| = 0$ most of the time. The only cases when we have $|\bar{v}_{\mathrm{III}}^{(s)}| = 1$ correspond to $\ell < s < i$. But this cannot occur because the sum over $i$ runs from $\ell + 1$ to $n$. In other words, the condition (5.13a) is always obeyed. Regarding the parameters $\bar{w}$, conditions (5.13b) and (5.13c) imply that $\bar{w}_{\mathrm{I}}^{(s)} = \{z, z_s\}$ for $0 \le s \le \ell$ and $\bar{w}_{\mathrm{I}}^{(s)} = \{z_s\}$ for $\ell + 1 \le s \le n - 1$. Then, we can replace the sum over partitions of $\bar{w}^{(s)}$ by a sum over partitions of $\bar{x}^{(s)} = \bar{u}^{(s)}$ for $0 \le s \le \ell$ and $\bar{x}^{(s)} = \{\bar{u}^{(s)}, z\}$ for $\ell + 1 \le s \le n - 1$. Moreover, looking at the cardinalities (3.1) for $\bar{w}_{\mathrm{I}}^{(s)}$, we see that we must take $i = \ell + 1$ to have cardinalities compatible with the sets $\bar{w}_{\mathrm{I}}^{(s)}$ given above.

From the subsets $\bar{w}_{\mathrm{I}}^{(s)}$ and $\bar{x}^{(s)} = \{\bar{w}_{\mathrm{II}}^{(s)}, \bar{w}_{\mathrm{III}}^{(s)}\}$

$$\bar{w}_{\mathrm{I}}^{(s)} = \begin{cases} \{z, z_s\}, & 0 \le s \le \ell, \\ \{z_s\}, & \ell + 1 \le s \le n - 1, \end{cases} \qquad \bar{x}^{(s)} = \begin{cases} \bar{u}^{(s)}, & 0 \le s \le \ell, \\ \{\bar{u}^{(s)}, z\}, & \ell + 1 \le s \le n - 1, \end{cases} \tag{5.14}$$

described above and the boundary subsets $\bar{w}_{\mathrm{I}}^{(n)} = \{z_n\}$ and $\bar{x}^{(n)} = \bar{w}_{\mathrm{III}}^{(n)} = \{z\}$ one can calculate

$$\Omega(\bar{w}_{\mathrm{I}}|\bar{w}_{\mathrm{II}})\,\Omega(\bar{w}_{\mathrm{I}}|\bar{w}_{\mathrm{III}}) = \Omega(\bar{w}_{\mathrm{I}}|\bar{x}) = \kappa \, \frac{h(z, \bar{u}^{(0)})}{g(z_1, \bar{u}^{(0)})} \, g(z, \bar{u}^{(\ell)})\, h(\bar{u}^{(\ell+1)}, z) \tag{5.15}$$

which proves the equality (5.9) from (3.3). The equalities (5.10) and (5.11) follows from the definitions of the functions $\Psi_{j,i}^{(\ell)}(\bar{v}, z)$ (3.16) and (3.17) since $\bar{v}_{\mathrm{III}}^{(s)} = \varnothing$ for all $s = 0, 1, \dots, n - 1$.

Altogether, we get

$$S(\bar{v}, z^{(\ell)}|\bar{u}) \simeq \prod_{s=0}^{n-1} \alpha_s(\bar{u}^{(s)}) \sum_{j=-n}^{\ell} \sum_{\text{part}} \Phi_{\ell+1,j}(\bar{x})\, \Psi_{j,\ell+1}^{(\ell)}(\bar{v}, z)\, Z(\bar{x}_{\mathrm{II}}|\bar{v}_{\mathrm{II}}) \,,$$

which leads to the relation (5.6). $\qquad\qquad\square$

**Proposition 5.3** *The highest coefficient also obeys the following recurrence relation*

$$Z(\bar{v}, z^{(\ell)}|\bar{u}) = \sum_{i=\ell+1}^{n} \sum_{\text{part}} \Phi_{i,\ell}(\bar{w})\, \Psi_{\ell,i}^{(\ell)}(\bar{v}, z)\, Z(\bar{v}_{\mathrm{II}}|\bar{w}_{\mathrm{II}}) \,, \tag{5.16}$$

*with for $\ell > 0$*

$$\Phi_{i,\ell}(\bar{w}) = \frac{g(z_1, \bar{u}^{(0)})}{h(z, \bar{u}^{(0)})} \, h(z, \bar{u}^{(\ell-1)})\, g(\bar{u}^{(\ell)}, z)\, \Omega(\bar{w}_{\mathrm{I}}|\bar{w}_{\mathrm{II}}) \,, \tag{5.17}$$

$$\Psi_{\ell,i}^{(\ell)}(\bar{v}, z) = \frac{\Omega(\bar{v}_{\mathrm{II}}|\bar{v}_{\mathrm{III}})}{g(z, \bar{v}^{(\ell-1)})\, h(z, \bar{v}^{(\ell)})\, h(\bar{v}^{(\ell)}, z)\, g(\bar{v}_{\mathrm{II}}^{(\ell+1)}, z)} \,, \tag{5.18}$$

*while for $\ell = 0$*

$$\Phi_{i,0}(\bar{w}) = -\, g(z, \bar{u}^{(0)})\, \Omega(\bar{w}_{\mathrm{I}}|\bar{w}_{\mathrm{II}}) \,, \tag{5.19}$$

$$\Psi_{0,i}^{(0)}(\bar{v}, z) = \frac{\Omega(\bar{v}_\mathrm{II}|\bar{v}_\mathrm{III})}{\mathfrak{f}(z_0, \bar{v}^{(0)})g(\bar{v}_\mathrm{II}^{(1)}, z)} \, . \tag{5.20}$$

*The sum is over partitions* $\bar{v}^{(s)} \vdash \{\bar{v}_\mathrm{II}^{(s)}, \bar{v}_\mathrm{III}^{(s)}\}$ *and* $\bar{w}^{(s)} \vdash \{\bar{w}_\mathrm{I}^{(s)}, \bar{w}_\mathrm{II}^{(s)}\}$ *with*

$$\bar{w}^{(s)} = \{\bar{u}^{(s)}, z, z_s\} \quad for \quad 0 \le s < \ell \, , \qquad \bar{w}^{(s)} = \{\bar{u}^{(s)}, z_s\} \quad for \quad \ell \le s \le n-1 \, . \tag{5.21}$$

*The cardinalities are given by*

$$for \quad 0 \le s \le \ell : \quad |\bar{v}_\mathrm{III}^{(s)}| = 0; \qquad for \quad \ell < s \le n-1 : \quad |\bar{v}_\mathrm{III}^{(s)}| = \begin{cases} 1, & s < i, \\ 0, & i \le s, \end{cases} \tag{5.22}$$

*and (for* $i \ge \ell + 1$*)*

$$\left|\bar{w}_\mathrm{I}^{(s)}\right| = \begin{cases} 2, & s < i, \\ 1, & i \le s, \end{cases} \qquad \bar{w}_\mathrm{III}^{(s)} = \begin{cases} \varnothing, & s < \ell, \\ \{z\}, & \ell \le s. \end{cases} \tag{5.23}$$

*Proof:* From (4.11), we have

$$S(\bar{v}, z^{(\ell)}|\bar{u}) \sim Z(\bar{v}, z^{(\ell)}|\bar{u}) \, \alpha_\ell(z) \prod_{s=0}^{n-1} \alpha_s(\bar{v}^{(s)}) \, ,$$

where $\sim$ stands for expressions modulo terms that contain at least one factor $\alpha_s(u_j^{(s)})$ for some $j$ and some $s$, or at least one factor $\alpha_s(z)$ for some $s \ne \ell$, or at least one factor[4] $\alpha_s(z_s)$ for some $s$. Starting from the recurrence relation (5.1) and using (4.11) for the scalar product $S(\bar{v}_\mathrm{II}|\bar{w}_\mathrm{II})$, we get

$$S(\bar{v}, z^{(\ell)}|\bar{u}) = \sum_{j=-n}^{\ell} \sum_{i=\ell+1}^{n} \sum_{\mathrm{part}} \frac{\left(\prod_{s=\ell+1}^{i-1} \alpha_s(\bar{v}_\mathrm{III}^{(s)})\right)\left(\prod_{s=0}^{n-1} \alpha_s(\bar{w}_\mathrm{III}^{(s)})\right)}{\prod_{s=\ell+1}^{n-1} \alpha_s(z)} \Phi_{ij}(\bar{w}) \, \Psi_{ji}^{(\ell)}(\bar{v}, z) S(\bar{v}_\mathrm{II}|\bar{w}_\mathrm{II})$$

$$\sim \sum_{j=-n}^{\ell} \sum_{i=\ell+1}^{n} \sum_{\mathrm{part}} \frac{\left(\prod_{s=\ell+1}^{i-1} \alpha_s(\bar{v}_\mathrm{III}^{(s)})\right)\left(\prod_{s=0}^{n-1} \alpha_s(\bar{w}_\mathrm{III}^{(s)})\right)}{\prod_{s=\ell+1}^{n-1} \alpha_s(z)} \Phi_{ij}(\bar{w}) \, \Psi_{ji}^{(\ell)}(\bar{v}, z) \prod_{s=0}^{n-1} \alpha_s(\bar{v}_\mathrm{II}^{(s)}) \, Z(\bar{v}_\mathrm{II}|\bar{w}_\mathrm{II}) \, .$$

This implies that we must have $\bar{w}_\mathrm{III}^{(s)} = \varnothing$ for $0 \le s < \ell$ and $\bar{w}_\mathrm{III}^{(s)} = \{z\}$ for $\ell \le s \le n-1$. Looking at the cardinalities (3.1), it implies that we must have $j \ge \ell$. Since the sum over $j$ runs up to $\ell$, it implies that $j = \ell$, so that

$$S(\bar{v}, z^{(\ell)}|\bar{u}) \sim \sum_{i=\ell+1}^{n} \sum_{part} \Phi_{i\ell}(\bar{w}) \, \Psi_{\ell i}^{(\ell)}(\bar{v}, z) \, Z(\bar{v}_\mathrm{II}|\bar{w}_\mathrm{II}) \, \alpha_\ell(z) \prod_{s=0}^{n-1} \alpha_s(\bar{v}^{(s)}) \, . \tag{5.24}$$

From (3.3) and (3.1), we obtain (5.17) and the cardinalities (5.23). Looking at the $\bar{v}$ part, since $j = \ell$, one sees that $|\bar{v}_\mathrm{I}^{(s)}| = 0$ for any $s$, and we get a partition $\bar{v}^{(s)} \vdash \{\bar{v}_\mathrm{II}^{(s)}, \bar{v}_\mathrm{III}^{(s)}\}$ with cardinalities (5.22), deduced from (3.20) for the specific values of $i$ and $j$. Then, from (3.16) and (3.17) we obtain (5.18) and (5.20). $\qquad\square$

---

[4]Factors $\alpha_s(z_s)$ do not appear in (4.1), so that this requirement is useless here. However, this exclusion is needed in the following.

## 5.2 Reduction to $Y(\mathfrak{gl}_n)$ models

For $\ell > 0$ the recurrence relations (5.6) and (5.16) may be compared with the recurrence relations for the highest coefficients obtained in the papers [7, 16]. In the first paper only the case $\ell = n - 1$ was considered, while in the second paper the general case of $1 \le \ell \le n - 1$ was investigated in the case of $U_q(\mathfrak{gl}_n)$ invariant models. These results can be easily translated to the case of $\mathfrak{gl}_n$ invariant models and compared with the results obtained above and specialized to the $\mathfrak{gl}_n$ case.

To performe this specialization we consider the above relations for the highest coefficient in the particular case $\bar{v}^{(0)} = \bar{u}^{(0)} = \varnothing$. In that case $\bar{v}_{\text{II}}^{(0)} = \varnothing$, which implies that $\bar{w}_{\text{II}}^{(0)} = \varnothing$ and the relations are valid for highest coefficients of $Y(\mathfrak{gl}_n)$ models. We detail them below. We will also show that they generalize some formulas already obtained in the context of $Y(\mathfrak{gl}_n)$ models.

**Proposition 5.4** *For $Y(\mathfrak{gl}_n)$ models, the highest coefficient obeys the recurrence relation for $\ell > 0$*

$$Z(\bar{u}|\bar{v}, z^{(\ell)}) = \sum_{j=1}^{\ell} \sum_{\text{part}} \Phi_{\ell+1,j}(\bar{x}) \, \Psi_{j,\ell+1}^{(\ell)}(\bar{v}, z) \, Z(\bar{x}_{\text{II}}|\bar{v}_{\text{II}}) \,, \tag{5.25}$$

*where*

$$\Phi_{\ell+1,j}(\bar{x}) = g(z, \bar{u}^{(\ell)}) \, h(\bar{u}^{(\ell+1)}, z) \prod_{s=j}^{n} \gamma(\bar{x}_{\text{II}}^{(s)}, \bar{x}_{\text{III}}^{(s)}) \frac{\prod_{s=j}^{n} h(\bar{x}_{\text{III}}^{(s)}, \bar{x}_{\text{II}}^{(s-1)})}{\prod_{s=j}^{n-2} g(\bar{x}_{\text{II}}^{(s+1)}, \bar{x}_{\text{III}}^{(s)})} \,, \tag{5.26}$$

$$\begin{aligned} \Psi_{j,\ell+1}^{(\ell)}(\bar{v}, z) = \Big( g(z, \bar{v}_{\text{II}}^{(\ell-1)}) \, h(z, \bar{v}^{(\ell)}) \, h(\bar{v}^{(\ell)}, z) \, g(\bar{v}^{(\ell+1)}, z) \Big)^{-1} \times \\ \times \prod_{s=j}^{\ell-1} \gamma(\bar{v}_{\text{I}}^{(s)}, \bar{v}_{\text{II}}^{(s)}) \frac{h(\bar{v}_{\text{II}}^{(s+1)}, \bar{v}_{\text{I}}^{(s)})}{g(\bar{v}_{\text{I}}^{(s+1)}, \bar{v}_{\text{II}}^{(s)})} \,. \end{aligned} \tag{5.27}$$

*The summation is over partitions of $\bar{x}^{(s)} \vdash \{\bar{x}_{\text{II}}^{(s)}, \bar{x}_{\text{III}}^{(s)}\}$ and $\bar{v}^{(s)} \vdash \{\bar{v}_{\text{I}}^{(s)}, \bar{v}_{\text{II}}^{(s)}\}$ with cardinalities given in (5.8). We recall that $\bar{x}^{(s)} = \bar{u}^{(s)}$ for $s \le \ell$ and $\bar{x}^{(s)} = \{\bar{u}^{(s)}, z\}$ for $s > \ell$.*

*Proof:* We consider the recurrence relation of the highest coefficient, equation (5.6), in the particular case $\bar{x}^{(0)} = \bar{v}^{(0)} = \bar{u}^{(0)} = \varnothing$, which implies that $\bar{x}_{\text{II}}^{(0)} = \bar{x}_{\text{III}}^{(0)} = \varnothing$. Since $\bar{x}_{\text{III}}^{(0)} = \varnothing$, the cardinalities (5.8) imply that we must take $j > 0$, and that there is no partition of $\bar{x}^{(s)}$ when $s < j$. Moreover, since $j \le \ell$ and $\bar{x}^{(s)} = \bar{u}^{(s)}$ when $s \le \ell$, in (5.9) we can split the product on $s$ in two parts. It leads to (5.26). To get (5.27), one notices that $|\bar{v}_{\text{I}}^{(s)}| = 0$ if $s < j$ or $s \ge \ell$. $\qquad\square$

**Proposition 5.5** *For $Y(\mathfrak{gl}_n)$ models and $\ell > 0$, the highest coefficient also obeys the recurrence relation*

$$Z(\bar{u}|\bar{v}, z^{(\ell)}) = \sum_{i=\ell+1}^{n} \sum_{\text{part}} \Phi_{i,\ell}(\bar{x}) \, \Psi_{\ell,i}^{(\ell)}(\bar{v}, z) \, Z(\bar{v}_{\text{II}}|\bar{x}_{\text{II}}) \,, \tag{5.28}$$

*where*

$$\Phi_{i,\ell}(\bar{x}) = h(z, \bar{u}^{(\ell-1)}) g(\bar{u}^{(\ell)}, z) \prod_{s=1}^{i-1} \gamma(\bar{x}_{\text{I}}^{(s)}, \bar{x}_{\text{II}}^{(s)}) \frac{\prod_{s=0}^{i-1} h(\bar{x}_{\text{II}}^{(s+1)}, \bar{x}_{\text{I}}^{(s)})}{\prod_{s=1}^{i-2} g(\bar{x}_{\text{I}}^{(s+1)}, \bar{x}_{\text{II}}^{(s)})} \tag{5.29}$$

$$\Psi_{\ell,i}^{(\ell)}(\bar{v},z) = \Big(g(z,\bar{v}^{(\ell-1)})\, h(z,\bar{v}^{(\ell)})\, h(\bar{v}^{(\ell)},z)\, g(\bar{v}_{\mathrm{II}}^{(\ell+1)},z)\Big)^{-1} \times$$

$$\times \prod_{s=\ell+1}^{i-1} \gamma_s(\bar{v}_{\mathrm{II}}^{(s)}, \bar{v}_{\mathrm{III}}^{(s)}) \frac{h(\bar{v}_{\mathrm{III}}^{(s)}, \bar{v}_{\mathrm{II}}^{(s-1)})}{g(\bar{v}_{\mathrm{II}}^{(s+1)}, \bar{v}_{\mathrm{III}}^{(s)})}. \tag{5.30}$$

*The sum runs over partitions* $\bar{x}^{(s)} \vdash \{\bar{x}_{\mathrm{I}}^{(s)}, \bar{x}_{\mathrm{II}}^{(s)}, \bar{x}_{\mathrm{III}}^{(s)}\}$ *and* $\bar{v}^{(s)} \vdash \{\bar{v}_{\mathrm{I}}^{(s)}, \bar{v}_{\mathrm{II}}^{(s)}, \bar{v}_{\mathrm{III}}^{(s)}\}$, *where* $\bar{x}^{(s)} = \bar{u}^{(s)}$ *for* $s < \ell$ *and* $\bar{x}^{(s)} = \{\bar{u}^{(s)}, z\}$ *for* $s \geq \ell$. *The cardinalities are given by*

$$\left|\bar{x}_{\mathrm{I}}^{(s)}\right| = \begin{cases} 1, & 0 < s < i, \\ 0, & \ell+1 \leq i \leq s, \end{cases} \qquad \bar{x}_{\mathrm{III}}^{(s)} = \begin{cases} \varnothing, & 0 < s < \ell, \\ \{z\}, & \ell \leq s, \end{cases}$$

$$\left|\bar{v}_{\mathrm{I}}^{(s)}\right| = 0, \quad \forall s, \qquad \left|\bar{v}_{\mathrm{III}}^{(s)}\right| = \begin{cases} 1, & \ell < s < i, \\ 0, & otherwise. \end{cases} \tag{5.31}$$

*Proof:* We consider the recurrence relation of the highest coefficient, equation (5.16), in the particular case $\bar{v}^{(0)} = \bar{u}^{(0)} = \varnothing$. In that case, $\bar{w}^{(0)} = \{z, z_0\}$ with $|\bar{w}_{\mathrm{I}}^{(0)}| = 2$, so that there is no partition for $\bar{w}^{(0)}$. Then, as a first step, we get

$$\Phi_{i\ell}(\bar{w}) = h(z, \bar{u}^{(\ell-1)}) g(\bar{u}^{(\ell)}, z) h(\bar{w}_{\mathrm{II}}^{(1)}, z) h(\bar{w}_{\mathrm{II}}^{(1)}, z_0) \prod_{s=1}^{n-1} \gamma_s(\bar{w}_{\mathrm{I}}^{(s)}, \bar{w}_{\mathrm{II}}^{(s)}) \frac{h(\bar{w}_{\mathrm{II}}^{(s+1)}, \bar{w}_{\mathrm{I}}^{(s)})}{g(\bar{w}_{\mathrm{I}}^{(s+1)}, \bar{w}_{\mathrm{II}}^{(s)})}. \tag{5.32}$$

The factor $h(\bar{w}_{\mathrm{II}}^{(1)}, z_0)$ implies that $z_1 \notin \bar{w}_{\mathrm{II}}^{(1)}$ so that $z_1 \in \bar{w}_{\mathrm{I}}^{(1)}$. Then due to the factor $h(\bar{w}_{\mathrm{II}}^{(s+1)}, \bar{w}_{\mathrm{I}}^{(s)})$, we get $z_s \in \bar{w}_{\mathrm{I}}^{(s)}$. Hence, we can replace the sum on partitions of $\bar{w}^{(s)} = \{\bar{u}^{(s)}, z, z_s\}$ by a partition on $\bar{x}^{(s)}$ with $\bar{x}^{(s)} = \bar{u}^{(s)}$ for $s < \ell$ and $\bar{x}^{(s)} = \{\bar{u}^{(s)}, z\}$ for $s \geq \ell$. The connexion between $\bar{w}^{(s)}$ and $\bar{x}^{(s)}$ is given by $\bar{w}_{\mathrm{I}}^{(s)} = \{\bar{x}_{\mathrm{I}}^{(s)}, z_s\}$, $\bar{w}_{\mathrm{II}}^{(s)} = \bar{x}_{\mathrm{II}}^{(s)}$ and $\bar{w}_{\mathrm{III}}^{(s)} = \bar{x}_{\mathrm{III}}^{(s)}$. It leads to (5.29). The cardinalities for $\bar{x}$ are deduced from those of $\bar{w}$, see (5.23). It allows to reduce the product on $s$ from $1 \leq s \leq n-1$ to $1 \leq s \leq i-1$.

In the same way, the cardinalities on $\bar{v}$ are deduced from (5.22) and allows to reduce the product on $s$. $\qquad\square$

The rational version of the BV scalar product computed in the paper [16] reads

$$\tilde{S}(\bar{v}|\bar{u}) = \tilde{\mathbb{C}}(\bar{v})\, \tilde{\mathbb{B}}(\bar{u}) = \sum_{\text{part}} \tilde{Z}(\bar{v}_{\mathrm{I}}|\bar{u}_{\mathrm{I}})\, \tilde{Z}(\bar{u}_{\mathrm{II}}|\bar{v}_{\mathrm{II}}) \prod_{s=1}^{n-1} \alpha_s(\bar{v}_{\mathrm{II}}^{(s)})^{-1}\, \alpha_s(\bar{u}_{\mathrm{I}}^{(s)})^{-1} \times$$

$$\times \prod_{s=1}^{n-1} \Big( f(\bar{v}_{\mathrm{II}}^{(s)}, \bar{v}_{\mathrm{I}}^{(s)})\, f(\bar{u}_{\mathrm{I}}^{(s)}, \bar{u}_{\mathrm{II}}^{(s)}) \Big) \prod_{s=1}^{n-2} \Big( f(\bar{v}_{\mathrm{II}}^{(s+1)}, \bar{v}_{\mathrm{I}}^{(s)})\, f(\bar{u}_{\mathrm{I}}^{(s+1)}, \bar{u}_{\mathrm{II}}^{(s)}) \Big)^{-1}. \tag{5.33}$$

In order to compare this scalar product with the scalar product (4.11) one has to renormalize the Bethe and dual Bethe vectors as follows

$$\mathbb{B}(\bar{u}) = \prod_{s=1}^{n-1} \alpha_s(\bar{u}^{(s)}) \prod_{s=1}^{n-2} h(\bar{u}^{(s+1)}, \bar{u}^{(s)}) \prod_{s=1}^{n-1} h(\bar{u}^{(s)}, \bar{u}^{(s)})^{-1}\, \tilde{\mathbb{B}}(\bar{u}),$$

$$\mathbb{C}(\bar{v}) = \prod_{s=1}^{n-1} \alpha_s(\bar{v}^{(s)}) \prod_{s=1}^{n-2} h(\bar{v}^{(s+1)}, \bar{v}^{(s)}) \prod_{s=1}^{n-1} h(\bar{v}^{(s)}, \bar{v}^{(s)})^{-1}\, \tilde{\mathbb{C}}(\bar{v}). \tag{5.34}$$

Comparing now the scalar product (4.11) for these Bethe vectors with the scalar product (5.33) one gets the relation between the highest coefficients $Z(\bar{v}|\bar{u})$ and $\tilde{Z}(\bar{v}|\bar{u})$

$$Z(\bar{u}|\bar{v}) = \tilde{Z}(\bar{u}|\bar{v}) \prod_{s=1}^{n-1} \frac{h(\bar{u}^{(s+1)}, \bar{u}^{(s)}) \, h(\bar{v}^{(s+1)}, \bar{v}^{(s)})}{h(\bar{u}^{(s)}, \bar{u}^{(s)}) \, h(\bar{v}^{(s)}, \bar{v}^{(s)})} \tag{5.35}$$

which allows to compare the recurrence relations obtained in [16] and specialized to the rational case with the relations (5.25) and (5.28) above.

After renaming the sets of Bethe parameters, one of the recurrence relation for the highest coefficient found in the paper [16] can be written in the following form

$$\tilde{Z}(\bar{u}|\bar{v}, z^{(\ell)}) = \frac{1}{f(z, \bar{v}^{(\ell-1)}) \, f(\bar{v}^{(\ell+1)}, z)} \sum_{j=1}^{\ell} \sum_{\text{part}} \prod_{s=j}^{\ell-1} \frac{g(\bar{v}_{\mathrm{I}}^{(s+1)}, \bar{v}_{\mathrm{I}}^{(s)}) \, f(\bar{v}_{\mathrm{I}}^{(s)}, \bar{v}_{\mathrm{II}}^{(s)})}{f(\bar{v}_{\mathrm{I}}^{(s)}, \bar{v}^{(s-1)})} \times$$

$$\times f(z, \bar{u}^{(\ell)}) \prod_{s=j}^{n-1} \frac{g(\bar{x}_{\mathrm{III}}^{(s+1)}, \bar{x}_{\mathrm{III}}^{(s)}) \, f(\bar{x}_{\mathrm{II}}^{(s)}, \bar{x}_{\mathrm{III}}^{(s)})}{f(\bar{x}^{(s+1)}, \bar{x}_{\mathrm{III}}^{(s)})} \times \tag{5.36}$$

$$\times \tilde{Z}\Big(\{\bar{x}^{(s)}\}_1^{j-1}, \{\bar{x}_{\mathrm{II}}^{(s)}\}_j^{n-1} | \{\bar{v}^{(s)}\}_1^{j-1}, \{\bar{v}_{\mathrm{II}}^{(s)}\}_j^{\ell-1}, \{\bar{v}^{(s)}\}_\ell^{n-1}\Big)$$

where in (5.36) $\bar{v}_{\mathrm{I}}^{(\ell)} \equiv \{z\}$, $\bar{v}_{\mathrm{II}}^{(\ell)} \equiv \bar{v}^{(\ell)}$ and the summation goes over partitions of the sets described in proposition 5.4. The recurrence relation (5.36) for $\ell = n - 1$ coincides with the relation (4.26) of [7].

Multiplying now both sides of the recurrence relation (5.36) by the product

$$\frac{h(z, \bar{v}^{(\ell-1)}) \, h(\bar{v}^{(\ell+1)}, z)}{h(z, \bar{v}^{(\ell)}) \, h(\bar{v}^{(\ell)}, z)} \prod_{s=1}^{n-1} \frac{h(\bar{v}^{(s+1)}, \bar{v}^{(s)})}{h(\bar{v}^{(s)}, \bar{v}^{(s)})} \prod_{s=1}^{n-1} \frac{h(\bar{u}^{(s+1)}, \bar{u}^{(s)})}{h(\bar{u}^{(s)}, \bar{u}^{(s)})} \tag{5.37}$$

and using the relation

$$\prod_{s=1}^{n-1} \frac{h(\bar{u}^{(s+1)}, \bar{u}^{(s)})}{h(\bar{u}^{(s)}, \bar{u}^{(s)})} = \frac{h(\bar{u}^{(\ell+1)}, z)}{h(z, \bar{u}^{(\ell)})} \prod_{s=1}^{n-1} \frac{h(\bar{x}^{(s+1)}, \bar{x}^{(s)})}{h(\bar{x}^{(s)}, \bar{x}^{(s)})} \tag{5.38}$$

which follows from definition of the sets $\bar{x}^{(s)}$, one gets the recurrence relations (5.25) with the functions $\Phi_{\ell+1,j}(\bar{w})$ and $\Psi_{j,\ell+1}^{(\ell)}(\bar{v}, z)$ given by equalities (5.26) and (5.27) respectively.

Analogously, renaming the sets of Bethe parameters in the second recurrence relation for the highest coefficient in [16], one can present it in the form

$$\tilde{Z}(\bar{v}, z^{(\ell)}|\bar{u}) = \frac{1}{f(z, \bar{v}^{(\ell-1)}) \, f(\bar{v}^{(\ell+1)}, z)} \sum_{i=\ell+1}^{n} \sum_{\text{part}} \prod_{s=\ell+1}^{i-1} \frac{g(\bar{v}_{\mathrm{III}}^{(s)}, \bar{v}_{\mathrm{III}}^{(s-1)}) \, f(\bar{v}_{\mathrm{II}}^{(s)}, \bar{v}_{\mathrm{III}}^{(s)})}{f(\bar{v}^{(s+1)}, \bar{v}_{\mathrm{III}}^{(s)})} \times$$

$$\times f(\bar{u}^{(\ell)}, z) \prod_{s=1}^{i-1} \frac{g(\bar{x}_{\mathrm{I}}^{(s)}, \bar{x}_{\mathrm{I}}^{(s-1)}) \, f(\bar{x}_{\mathrm{I}}^{(s)}, \bar{x}_{\mathrm{II}}^{(s)})}{f(\bar{x}_{\mathrm{I}}^{(s)}, \bar{x}^{(s-1)})} \times \tag{5.39}$$

$$\times \tilde{Z}\Big(\{\bar{v}^{(s)}\}_1^{\ell}, \{\bar{v}_{\mathrm{II}}^{(s)}\}_{\ell+1}^{i-1}, \{\bar{v}^{(s)}\}_i^{n-1} | \{\bar{x}^{(s)}\}_1^{i-1}, \{\bar{x}_{\mathrm{II}}^{(s)}\}_i^{n-1}\Big)$$

where again in (5.39) $\bar{v}_{\mathrm{I}}^{(\ell)} \equiv \{z\}$, $\bar{v}_{\mathrm{II}}^{(\ell)} \equiv \bar{v}^{(\ell)}$ and the summation goes over partitions of the sets described in proposition 5.5. Multiplying both sides of the recurrence equality (5.39) by the product (5.37) and using the relation

$$\prod_{s=1}^{n-1} \frac{h(\bar{u}^{(s+1)}, \bar{u}^{(s)})}{h(\bar{u}^{(s)}, \bar{u}^{(s)})} = \frac{h(z, \bar{u}^{(\ell-1)})}{h(\bar{u}^{(\ell)}, z)} \prod_{s=1}^{n-1} \frac{h(\bar{x}^{(s+1)}, \bar{x}^{(s)})}{h(\bar{x}^{(s)}, \bar{x}^{(s)})} \tag{5.40}$$

which follows from the definition of the sets $\bar{x}^{(s)}$ in proposition 5.5 we find the recurrence relation (5.28) with the functions $\Phi_{i,\ell}(\bar{w})$ and $\Psi_{\ell,i}^{(\ell)}(\bar{v}, z)$ defined by the formulas (5.29) and (5.30) respectively.

## 5.3 Recurrence relations for $Y(\mathfrak{gl}_2)$ and $Y(\mathfrak{o}_3)$ models

It was checked in [15] that the action formulas (3.6) are valid also in the case $n = 1$. Similarly, it was verified in [17] that the recurrence relations (3.14) and correspondingly (3.19) coincide with the recurrence relations obtained in [18] for $Y(\mathfrak{o}_3)$ models.

On the other hand due to the isomorphism between Yangians $Y(\mathfrak{o}_3)$ and $Y(\mathfrak{gl}_2)$ the highest coefficients in $\mathfrak{o}_3$ invariant models should be related to the highest coefficients in $\mathfrak{gl}_2$ invariant models. To describe this relation we first consider the recurrence relations (5.25) and (5.28) in the case $n = 2$, $\ell = 1$ and provide their solutions in term of the Izergin determinant.

**Izergin determinant.** For a set $\bar{u}$ of cardinality $|\bar{u}| = r$ we introduce the following triangular products of $g$ functions

$$\delta_g(\bar{u}) = \prod_{1 \le a < b \le r} g(u_b, u_a), \qquad \delta'_g(\bar{u}) = \prod_{1 \le a < b \le r} g(u_a, u_b). \tag{5.41}$$

The Izergin determinant is the rational function $K_r^{(c)}(\bar{v}|\bar{u})$ depending on two sets of formal variables $\bar{v}$ and $\bar{u}$ of the same cardinality $|\bar{v}| = |\bar{u}| = r$

$$K_r^{(c)}(\bar{v}|\bar{u}) = h(\bar{v}, \bar{u}) \, \delta_g(\bar{u}) \, \delta'_g(\bar{v}) \, \det \left| \frac{g(v_a, u_b)}{h(v_a, u_b)} \right|_{a,b=1,\dots,r}. \tag{5.42}$$

It satisfies different properties described in the book [23]. Its dependence on the parameter $c$ occurs through the functions $g(u, v)$ and $h(u, v)$. In particular, the Izergin determinant satisfies recurrence relations which can be written in the form

$$K_r^{(c)}(\bar{v}, z|\bar{u}) = f(z, \bar{u}) \sum_{\text{part}} \frac{f(\bar{u}_{\mathrm{II}}, \bar{u}_{\mathrm{I}})}{h(z, \bar{u}_{\mathrm{I}})} \, K_{r-1}^{(c)}(\bar{v}|\bar{u}_{\mathrm{II}}),$$

$$K_r^{(c)}(\bar{u}|\bar{v}, z) = f(\bar{u}, z) \sum_{\text{part}} \frac{f(\bar{u}_{\mathrm{I}}, \bar{u}_{\mathrm{II}})}{h(\bar{u}_{\mathrm{I}}, z)} \, K_{r-1}^{(c)}(\bar{u}_{\mathrm{II}}|\bar{v}) \tag{5.43}$$

for any sets $\bar{u}$ and $\bar{v}$ with cardinalities $|\bar{u}| = |\bar{v}| + 1$. In (5.43) the summation goes over the partitions $\bar{u} \vdash \{\bar{u}_{\mathrm{I}}, \bar{u}_{\mathrm{II}}\}$ such that the cardinality $|\bar{u}_{\mathrm{I}}| = 1$.

**Highest coefficients for $\mathfrak{gl}_2$ invariant models.** The recurrence relations (5.25) and (5.28) for the highest coefficients $Z(\bar{u}|\bar{v}, z)$ and $Z(\bar{v}, z|\bar{u})$ in $\mathfrak{gl}_2$ invariant models takes the form

$$Z(\bar{u}|\bar{v}, z) = \frac{f(z, \bar{u})}{h(z, \bar{v}) \, h(\bar{v}, z)} \sum_{\text{part}} \frac{f(\bar{u}_{\mathrm{II}}, \bar{u}_{\mathrm{III}})}{h(z, \bar{u}_{\mathrm{III}})} \frac{Z(\bar{u}_{\mathrm{II}}|\bar{v})}{h(\bar{u}_{\mathrm{II}}, \bar{u}_{\mathrm{III}}) \, h(\bar{u}_{\mathrm{III}}, \bar{u}_{\mathrm{II}})} \tag{5.44}$$

and

$$Z(\bar{v}, z|\bar{u}) = \frac{f(\bar{u}, z)}{h(z, \bar{v}) \, h(\bar{v}, z)} \sum_{\text{part}} \frac{f(\bar{u}_{\mathrm{I}}, \bar{u}_{\mathrm{II}})}{h(\bar{u}_{\mathrm{I}}, z)} \frac{Z(\bar{v}|\bar{u}_{\mathrm{II}})}{h(\bar{u}_{\mathrm{I}}, \bar{u}_{\mathrm{II}}) \, h(\bar{u}_{\mathrm{II}}, \bar{u}_{\mathrm{I}})} \tag{5.45}$$

respectively. Comparing these equalities with the recurrence relations for the Izergin determinant (5.43) we conclude that they can be solved as follows

$$Z(\bar{u}|\bar{v}) = \frac{K_r^{(c)}(\bar{v}|\bar{u})}{h(\bar{u}, \bar{u}) \, h(\bar{v}, \bar{v})}. \tag{5.46}$$

Note, that for the 'old' normalization of the Bethe vectors $\tilde{\mathbb{B}}(\bar{u})$ and $\tilde{\mathbb{C}}(\bar{v})$ the highest coefficient $\tilde{Z}(\bar{u}|\bar{v})$ in $Y(\mathfrak{gl}_2)$ based model coincides with the Izergin determinant $K_r^{(c)}(\bar{v}|\bar{u})$.

**Highest coefficients for $\mathfrak{o}_3$ invariant model.** Let us write explicitly the recurrence relations for the highest coefficients in $Y(\mathfrak{o}_3)$ models. In that case we have $n = 1$, $\ell = 0$, $\bar{u} \equiv \bar{u}^{(0)}$, $\bar{v} \equiv \bar{v}^{(0)}$ with cardinalities $|\bar{v}| + 1 = |\bar{u}| = r$.

The recurrence relation (5.6) given by the proposition 5.2 takes the form

$$Z(\bar{u}|\bar{v}, z) = \sum_{j=-1}^{0} \sum_{\text{part}} \Phi_{1,j}(\bar{u}) \, \Psi_{j,1}^{(0)}(\bar{v}, z) \; Z(\bar{u}_{\text{II}}|\bar{v}_{\text{II}}) \,, \tag{5.47}$$

where

$$\Phi_{1,j}(\bar{u}) \, \Psi_{j,1}^{(0)}(\bar{v}, z) = \left( \frac{f(z, \bar{u}) \, \mathfrak{f}(\bar{u}_{\text{II}}, \bar{u}_{\text{III}})}{h(z, \bar{u}_{\text{III}})} \right) \cdot \left( \frac{g(z_0, \bar{v}_{\text{I}}) \, \mathfrak{f}(\bar{v}_{\text{I}}, \bar{v}_{\text{II}})}{\mathfrak{f}(z_0, \bar{v})} \right) . \tag{5.48}$$

Using (5.48) the recurrence relation (5.47) takes the form

$$\begin{aligned}
Z(\bar{u}|\bar{v}, z) = &\frac{f(z, \bar{u})}{\mathfrak{f}(z_0, \bar{v})} \sum_{\text{part}} \frac{\mathfrak{f}(\bar{u}_{\text{II}}, \bar{u}_{\text{III}})}{h(z, \bar{u}_{\text{III}})} \; Z(\bar{u}_{\text{II}}|\bar{v}) + \\
&+ \frac{f(z, \bar{u})}{\mathfrak{f}(z_0, \bar{v})} \sum_{\text{part}} g(z_0, \bar{v}_{\text{I}}) \, \mathfrak{f}(\bar{v}_{\text{I}}, \bar{v}_{\text{II}}) \, \frac{\mathfrak{f}(\bar{u}_{\text{II}}, \bar{u}_{\text{III}})}{h(z, \bar{u}_{\text{III}})} \; Z(\bar{u}_{\text{II}}|\bar{v}_{\text{II}}) \,,
\end{aligned} \tag{5.49}$$

where in the first line of (5.49) sum goes over partitions of the set $\bar{u} \vdash \{\bar{u}_{\text{II}}, \bar{u}_{\text{III}}\}$ with cardinalities $|\bar{u}_{\text{III}}| = 1$, and in the second line of (5.49) sum goes over partitions of the sets $\bar{v} \vdash \{\bar{v}_{\text{I}}, \bar{v}_{\text{II}}\}$ and $\bar{u} \vdash \{\bar{u}_{\text{II}}, \bar{u}_{\text{III}}\}$ with cardinalities $|\bar{v}_{\text{I}}| = 1$, $|\bar{u}_{\text{III}}| = 2$.

Analogously one can present the recurrence relation (5.16) given in the proposition 5.3 for the case $n = 1$ and $\ell = 0$ in the form

$$Z(\bar{v}, z|\bar{u}) = \frac{\mathfrak{f}(\bar{u}, z)}{\mathfrak{f}(z_0, \bar{v})} \sum_{\text{part}} g(z_1, \bar{w}_{\text{I}}) \, \mathfrak{f}(\bar{w}_{\text{I}}, \bar{w}_{\text{II}}) \; Z(\bar{v}|\bar{w}_{\text{II}}) \tag{5.50}$$

and sum in (5.50) goes according to (5.22) and (5.23) which tell that there is no partition of the set $\bar{v}$ and the set $\bar{w} = \{\bar{u}, z_0\} = \{\bar{w}_{\text{I}}, \bar{w}_{\text{II}}\}$ (5.21) is parted with cardinality $|\bar{w}_{\text{I}}| = 2$.

**Proposition 5.6** *Both recurrence relations (5.49) and (5.50) are satisfied by the following expression for the highest coefficient of the scalar product in $Y(\mathfrak{o}_3)$ integrable models*

$$Z(\bar{u}|\bar{v}) = 2^{|\bar{u}|} \; K_r^{(c/2)}(\bar{v}|\bar{u}) \,. \tag{5.51}$$

*Proof:* The explicit proof of this proposition requires use of the summation identity for the Izergin's determinant which can be found in the book [23]. □

As it was expected from the isomorphism between $Y(\mathfrak{o}_3)$ and $Y(\mathfrak{gl}_2)$ at $c \to c/2$ the highest coefficients in these models are proportional to the Izergin's determinants with the parameters $c/2$ and $c$ respectively.

# 6 Analytical properties of the highest coefficients

Another type of recurrence relations can be obtained when considering the limit $v_j^{(s)} \to u_j^{(s)}$ for some $j$ and $s$. It provides the pole structure of the highest coefficient (HC) when the Bethe parameters of the Bethe vector and the dual Bethe vector coincide.

**Proposition 6.1** *For given fixed $p = 0, 1, ..., n-1$ and $k = 1, 2, ..., r_p$, in the limit $v_k^{(p)} \to u_k^{(p)}$ we have the equivalence*

$$Z(\bar{u}|\bar{v}) \sim g(u_k^{(p)}, v_k^{(p)}) \, A_k^{(p)}(\bar{u}|\bar{v}) \, Z(\mathring{\bar{u}}|\mathring{\bar{v}}) + reg.,$$

$$A_k^{(p)}(\bar{u}|\bar{v}) = \gamma_p(u_k^{(p)}, \bar{u}_k^{(p)}) \, \gamma_p(\bar{v}_k^{(p)}, v_k^{(p)}) \, \frac{h(\bar{u}^{(p+1)}, u_k^{(p)}) \, h(v_k^{(p)}, \bar{v}^{(p-1)})}{g(u_k^{(p)}, \bar{u}^{(p-1)}) \, g(\bar{v}^{(p+1)}, v_k^{(p)})} \tag{6.1}$$

$$= \Omega(u_k^{(p)}|\mathring{\bar{u}}) \Omega(\mathring{\bar{v}}|v_k^{(p)}),$$

*where $\mathring{\bar{u}} = \bar{u} \setminus \{u_k^{(p)}\}$, $\mathring{\bar{v}} = \bar{v} \setminus \{v_k^{(p)}\}$ and reg. denotes terms that are regular in the limit $v_k^{(p)} \to u_k^{(p)}$. By convention $\bar{u}^{(-1)} = \bar{v}^{(-1)} = \bar{u}^{(n)} = \bar{v}^{(n)} = \varnothing$.*

*Proof:* We prove the property through a recursion on $n$, the rank of $\mathfrak{o}_{2n+1}$.

First we note that for $n = 1$, the generalized models for $Y(\mathfrak{o}_3)$ are equivalent to those for $Y(\mathfrak{gl}_2)$, see [18]. One can deduce the scalar products as

$$S_{\mathfrak{o}_3}(\bar{v}|\bar{u}) = 2^{|\bar{u}|} \left( h(\bar{u}, \bar{u}) \, h(\bar{v}, \bar{v}) \, S_{\mathfrak{gl}_2}(\bar{v}|\bar{u}) \right)_{c \to c/2}, \tag{6.2}$$

which was shown in the previous section on the level of the HC. This implies that all the properties, which have already been proved in the $Y(\mathfrak{gl}_2)$ context, are valid for $n = 1$. In particular, the residue property is already proven for $n = 1$.

**Remark 6.1** *Assuming that $\bar{u}^{(0)} = \bar{v}^{(0)} = \varnothing$, we may also compare the property (6.1) with the Proposition 3.1 of [8]. Using the notation of the present article, we rewrite the equation (3.17) of [8] as*

$$\tilde{Z}(\bar{u}|\bar{v})\big|_{u_k^{(p)} \to v_k^{(p)}} = g(v_k^{(p)}, u_k^{(p)}) \, \frac{f(\bar{v}_k^{(p)}, v_k^{(p)}) f(u_k^{(p)}, \bar{u}_k^{(p)})}{f(\bar{v}^{(p+1)}, v_k^{(p)}) f(u_k^{(p)}, \bar{u}^{(p-1)})} \, \tilde{Z}(\mathring{\bar{u}}|\mathring{\bar{v}}) + reg. \tag{6.3}$$

*To do the comparison, we need to use the formula (5.35) which describes the relation between highest coefficients for the scalar product of the Bethe vectors with the normalization used in this paper and the same objects for the normalization used in [8]. This difference in normalization produces an additional factor for the residue property of the highest coefficients*

$$\Xi_k^{(p)}(\bar{u}, \bar{v}) = -\prod_{s=1}^{n-1} \frac{h(\bar{u}^{(s)}, \bar{u}^{(s)})}{h(\bar{u}^{(s+1)}, \bar{u}^{(s)})} \frac{h(\bar{v}^{(s)}, \bar{v}^{(s)})}{h(\bar{v}^{(s+1)}, \bar{v}^{(s)})} \frac{h(\mathring{\bar{u}}^{(s+1)}, \mathring{\bar{u}}^{(s)})}{h(\mathring{\bar{u}}^{(s)}, \mathring{\bar{u}}^{(s)})} \frac{h(\mathring{\bar{v}}^{(s+1)}, \mathring{\bar{u}}^{(v)})}{h(\mathring{\bar{v}}^{(s)}, \mathring{\bar{v}}^{(s)})}$$

$$= -\frac{h(\bar{u}_k^{(p)}, u_k^{(p)}) h(u_k^{(p)}, \bar{u}_k^{(p)})}{h(\bar{u}^{(p+1)}, u_k^{(p)}) h(u_k^{(p)}, \bar{u}^{(p-1)})} \frac{h(\bar{v}_k^{(p)}, v_k^{(p)}) h(v_k^{(p)}, \bar{v}_k^{(p)})}{h(\bar{v}^{(p+1)}, v_k^{(p)}) h(v_k^{(p)}, \bar{v}^{(p-1)})}.$$

*Using the expression (6.1) for $A_k^{(p)}(\bar{u}|\bar{v})$, it is easy to see that the product $\Xi_k^{(p)}(\bar{u}, \bar{v}) A_k^{(p)}(\bar{u}|\bar{v})$ just reproduces the factor in (6.3).*

Now, suppose that the property is true for $Y(\mathfrak{o}_{2m+1})$ models with $m < n$, and consider the HC for $Y(\mathfrak{o}_{2n+1})$ models. We show that the relation (6.1) is valid for $Y(\mathfrak{o}_{2n+1})$ models through a recurrence on $r_{n-1} = |\bar{u}^{(n-1)}|$. When $r_{n-1} = 0$, then we are back to a highest coefficient of $Y(\mathfrak{o}_{2n-1})$ models, and the relation is true by the induction hypothesis on $n$. We suppose now that it is true for $r_{n-1} < r$ and consider a highest coefficient with $|\bar{u}^{(n-1)}| = r$.

We first look at the case $p < n - 1$, with $v_k^{(p)} \to u_k^{(p)}$. We consider the recurrence relation (5.6) with $\ell = n - 1$. Obviously, poles corresponding to the limit $v_k^{(p)} \to u_k^{(p)}$ may appear only in the highest coefficient $Z(\bar{u}_{\mathrm{II}}|\bar{v}_{\mathrm{II}})$ when $v_k^{(p)} \in \bar{v}_{\mathrm{II}}^{(p)}$ and $u_k^{(p)} \in \bar{u}_{\mathrm{II}}^{(p)}$. In that case, we can use the induction hypothesis to get

$$Z(\bar{u}_{\mathrm{II}}|\bar{v}_{\mathrm{II}}) \sim g(u_k^{(p)}, v_k^{(p)}) A_k^{(p)}(\bar{u}_{\mathrm{II}}|\bar{v}_{\mathrm{II}}) Z(\mathring{\bar{u}}_{\mathrm{II}}|\mathring{\bar{v}}_{\mathrm{II}}) + reg. \, ,$$

where $\bar{v}_{\mathrm{I}}^{(p)} = \mathring{\bar{v}}_{\mathrm{I}}^{(p)}$, $\bar{v}_{\mathrm{II}}^{(p)} = \{\mathring{\bar{v}}_{\mathrm{II}}^{(p)}, v_k^{(p)}\}$, with $\mathring{\bar{v}} = \bar{v} \setminus \{v_k^{(p)}\}$ and similarly for $\bar{u}$. We have also for $0 \le p \le n - 2$

$$\Psi_{j,n}^{(n-1)}(\bar{v}, z) = \gamma_p(\bar{v}_{\mathrm{I}}^{(p)}, v_k^{(p)}) \frac{h(v_k^{(p)}, \bar{v}_{\mathrm{I}}^{(p-1)})}{g(\bar{v}_{\mathrm{I}}^{(p+1)}, v_k^{(p)})} \frac{\Psi_{j,n}(\mathring{\bar{v}}, z)}{\left(g(z, v_k^{(n-2)})\right)^{\delta_{p,n-2}}} \, , \tag{6.4}$$

$$\Phi_{n,j}(\bar{u}, z) = \gamma_p(u_k^{(p)}, \bar{u}_{\mathrm{III}}^{(p)}) \frac{h(\bar{u}_{\mathrm{III}}^{(p+1)}, u_k^{(p)})}{g(u_k^{(p)}, \bar{u}_{\mathrm{III}}^{(p-1)})} \Phi_{n,j}(\mathring{\bar{u}}, z) \, , \tag{6.5}$$

with the convention that $\bar{v}^{(-1)} = \bar{u}^{(-1)} = \varnothing$. This leads to the following formula

$$Z(\bar{u}|\bar{v}^{(0)}, \ldots, \{\bar{v}^{(n-1)}, z\}) = g(u_k^{(p)}, v_k^{(p)}) \sum_{j=-n}^{n-1} \sum_{\mathrm{part}} B_{p,k}^{j,n-1}(\bar{u}, \bar{v}, z) A_k^{(p)}(\bar{u}_{\mathrm{II}}|\bar{v}_{\mathrm{II}})$$
$$\times \Phi_{n,j}(\mathring{\bar{u}}, z) \Psi_{j,n}^{(n-1)}(\mathring{\bar{v}}, z) Z(\mathring{\bar{u}}_{\mathrm{II}}|\mathring{\bar{v}}_{\mathrm{II}}) + reg. \, ,$$

where $B_{p,k}^{j,n-1}(\bar{u}, \bar{v}, z)$ is deduced from formulas (6.4) and (6.5), and $A_k^{(p)}(\bar{u}_{\mathrm{II}}|\bar{v}_{\mathrm{II}})$ takes the form (6.1) by the induction hypothesis on $r_{n-1}$.

Now, looking at the expression of $B_{p,k}^{j,n-1}(\bar{u}, \bar{v}, z)$ and $A_k^{(p)}(\bar{u}_{\mathrm{II}}|\bar{v}_{\mathrm{II}})$, it is easy to see that

$$B_{p,k}^{j,n-1}(\bar{u}, \bar{v}, z) A_k^{(p)}(\bar{u}_{\mathrm{II}}|\bar{v}_{\mathrm{II}}) = A_k^{(p)}(\bar{u}|\{\bar{v}, z^{(n-1)}\}) \, . \tag{6.6}$$

When doing the calculation, one has to single out the case $p = n - 2$, because it makes appear $z$, as shown in (6.4). Note that the r.h.s. in (6.6) does not depend on the partition, nor on $j$. Then, since

$$\sum_{j=-n}^{n-1} \sum_{\mathrm{part}} \Phi_{n,j}(\mathring{\bar{u}}, z) \Psi_{j,n}^{(n-1)}(\mathring{\bar{v}}, z) Z(\mathring{\bar{u}}_{\mathrm{II}}|\mathring{\bar{v}}_{\mathrm{II}}) = Z(\mathring{\bar{u}}|\mathring{\bar{v}}^{(0)}, \ldots, \{\mathring{\bar{v}}^{(n-1)}, z\})$$

we get the expression for $r_{n-1} = r$. This ends the recurrence for $p < n - 1$.

We now look at the case $p = n - 1$. If $|\bar{u}^{(0)}| = |\bar{v}^{(0)}| = 0$, then $Z(\bar{u}|\bar{v}^{(1)}, \ldots, \bar{v}^{(n-1)})$ corresponds to a $Y(\mathfrak{gl}_n)$ model, for which the residue property holds. The exact comparison with the results (3.17) of the paper [8] was presented in the remark 6.1.

For $|\bar{u}^{(0)}| = |\bar{v}^{(0)}| > 0$, we consider the recurrence relation (5.6) with $\ell = 0$, in the limit $v_k^{(p)} \to u_k^{(p)}$, with $p > 1$. By the recurrence hypothesis, we have

$$Z(\bar{u}_{\mathrm{II}}|\bar{v}_{\mathrm{II}}) \sim g(u_k^{(p)}, v_k^{(p)}) A_k^{(p)}(\bar{u}_{\mathrm{II}}|\bar{v}_{\mathrm{II}}) Z(\mathring{\bar{u}}_{\mathrm{II}}|\mathring{\bar{v}}_{\mathrm{II}}) + reg.$$

One can also compute

$$\Psi_{j,1}^{(0)}(\bar{v}, z) = \gamma_p(\bar{v}_{\mathrm{I}}^{(p)}, v_k^{(p)}) \frac{h(v_k^{(p)}, \bar{v}_{\mathrm{I}}^{(p-1)})}{g(\bar{v}_{\mathrm{I}}^{(p+1)}, v_k^{(p)})} \Psi_{j,1}^{(0)}(\mathring{\bar{v}}, z),$$

$$\Phi_{1,j}(\bar{u}, z) = \gamma_p(u_k^{(p)}, \bar{u}_{\mathrm{III}}^{(p)}) \frac{h(\bar{u}_{\mathrm{III}}^{(p+1)}, u_k^{(p)})}{g(u_k^{(p)}, \bar{u}_{\mathrm{III}}^{(p-1)})} \Phi_{1,j}(\mathring{\bar{u}}, z)$$

leading to

$$B_{p,k}^{j,0}(\bar{u},\bar{v},z) = \gamma_p(\bar{v}_{\mathrm{I}}^{(p)},v_k^{(p)}) \frac{h(v_k^{(p)},\bar{v}_{\mathrm{I}}^{(p-1)})}{g(\bar{v}_{\mathrm{I}}^{(p+1)},v_k^{(p)})} \gamma_p(u_k^{(p)},\bar{u}_{\mathrm{III}}^{(p)}) \frac{h(\bar{u}_{\mathrm{III}}^{(p+1)},u_k^{(p)})}{g(u_k^{(p)},\bar{u}_{\mathrm{III}}^{(p-1)})}.$$

Once again we get

$$B_{p,k}^{j,0}(\bar{u},\bar{v},z)\,A_k^{(p)}(\bar{u}_{\mathrm{II}}|\bar{v}_{\mathrm{II}}) = A_k^{(p)}(\bar{u}|\{\bar{v},z^{(0)}\})\,. \tag{6.7}$$

It proves the identity (6.1) for $r_{n-1} = r$ and $1 < p \leq n-1$ and concludes the induction proof. $\qquad\square$

We checked that using the recurrence relation (5.16) instead of (5.6) leads to the same residue formula (6.1). As in the proof above, for a given color $p$, one can consider any recurrence relation (5.16) with $p \neq \ell-1, \ell, \ell-2$ to do the proof.

## 6.1 Residue formula for the scalar product

From the residue for the highest coefficient, we can deduce a similar relation for the scalar product. We consider $S(\bar{u}|\bar{v})$ in the limit $u_k^{(p)} \to v_k^{(p)}$. First, we prove that this limit is non-singular. Then, we show that the coefficient of the derivative of $\alpha_p(u_k^{(p)})$ can be interpreted as a scalar product of a similar type. Starting from the sum formula (4.11), the terms in which $u_k^{(p)} \in \bar{u}_{\mathrm{I}}^{(p)}$ and $v_k^{(p)} \in \bar{v}_{\mathrm{II}}^{(p)}$ or alternatively $u_k^{(p)} \in \bar{u}_{\mathrm{II}}^{(p)}$ and $v_k^{(p)} \in \bar{v}_{\mathrm{I}}^{(p)}$ have no singularities in the limit $u_k^{(p)} \to v_k^{(p)}$. The two contributions with $u_k^{(p)}$ (and $v_k^{(p)}$) in the subsets $\bar{u}_{\mathrm{I}}^{(p)}$ (and $\bar{v}_{\mathrm{I}}^{(p)}$) or in the subsets $\bar{u}_{\mathrm{II}}^{(p)}$ (and $\bar{v}_{\mathrm{II}}^{(p)}$) have a spurious singularity that needs to be dealt with. In the first case, applying the residue formula for the highest coefficient (6.1), we get:

$$\begin{aligned} S_1(\bar{v}|\bar{u}) = \sum_{\text{part}} & g(v_k^{(p)},u_k^{(p)})\alpha_p(v_k^{(p)})A_k^{(p)}(\bar{v}_{\mathrm{I}}|\bar{u}_{\mathrm{I}})\Omega(\bar{u}_{\mathrm{I}}|\bar{u}_{\mathrm{II}})\Omega(\bar{v}_{\mathrm{II}}|\bar{v}_{\mathrm{I}}) \\ & \times Z(\mathring{\bar{v}}_{\mathrm{I}}|\mathring{\bar{u}}_{\mathrm{I}})Z(\bar{u}_{\mathrm{II}}|\bar{v}_{\mathrm{II}})\prod_{s=0}^{n-1}\alpha_s(\mathring{\bar{v}}_{\mathrm{I}}^{(s)})\alpha_s(\bar{u}_{\mathrm{II}}^{(s)}) + reg. \,, \end{aligned} \tag{6.8}$$

where $\mathring{\bar{u}} = \bar{u} \setminus \{u_k^{(p)}\}$ and $\mathring{\bar{v}} = \bar{v} \setminus \{v_k^{(p)}\}$. In the second case we get

$$\begin{aligned} S_2(\bar{v}|\bar{u}) = \sum_{\text{part}} & g(u_k^{(p)},v_k^{(p)})\alpha_s(u_k^{(p)})A_k^{(p)}(\bar{u}_{\mathrm{II}}|\bar{v}_{\mathrm{II}})\Omega(\bar{u}_{\mathrm{I}}|\bar{u}_{\mathrm{II}})\Omega(\bar{v}_{\mathrm{II}}|\bar{v}_{\mathrm{I}}) \\ & \times Z(\bar{v}_{\mathrm{I}}|\bar{u}_{\mathrm{I}})Z(\mathring{\bar{u}}_{\mathrm{II}}|\mathring{\bar{v}}_{\mathrm{II}})\prod_{s=0}^{n-1}\alpha_s(\bar{v}_{\mathrm{I}}^{(s)})\alpha_s(\mathring{\bar{u}}_{\mathrm{II}}^{(s)}) + reg. \end{aligned} \tag{6.9}$$

Using the expression (6.1) of $A_k^{(p)}$, we get

$$\begin{aligned} S_1(\bar{v}|\bar{u}) =& g(v_k^{(p)},u_k^{(p)})\alpha_p(v_k^{(p)})\sum_{\text{part}}\Omega(v_k^{(p)}|\mathring{\bar{v}}_{\mathrm{I}})\Omega(\bar{v}_{\mathrm{II}}|u_k^{(p)})\Omega(\mathring{\bar{u}}_{\mathrm{I}}|u_k^{(p)})\Omega(u_k^{(p)}|\bar{u}_{\mathrm{II}}) \\ & \times \Omega(\bar{v}_{\mathrm{II}}|\mathring{\bar{v}}_{\mathrm{I}})\Omega(\mathring{\bar{u}}_{\mathrm{I}}|\bar{u}_{\mathrm{II}})Z(\mathring{\bar{v}}_{\mathrm{I}}|\mathring{\bar{u}}_{\mathrm{I}})Z(\bar{u}_{\mathrm{II}}|\bar{v}_{\mathrm{II}})\prod_{s=0}^{n-1}\alpha_s(\mathring{\bar{v}}_{\mathrm{I}}^{(s)})\alpha_s(\bar{u}_{\mathrm{II}}^{(s)}) + reg. \,, \\ S_2(\bar{v}|\bar{u}) =& g(u_k^{(p)},v_k^{(p)})\alpha_s(u_k^{(p)})\sum_{\text{part}}\Omega(v_k^{(p)}|\bar{v}_{\mathrm{I}})\Omega(\mathring{\bar{v}}_{\mathrm{II}}|v_k^{(p)})\Omega(\bar{u}_{\mathrm{I}}|u_k^{(p)})\Omega(u_k^{(p)}|\mathring{\bar{u}}_{\mathrm{II}}) \\ & \times \Omega(\mathring{\bar{v}}_{\mathrm{II}}|\bar{v}_{\mathrm{I}})\Omega(\bar{u}_{\mathrm{I}}|\mathring{\bar{u}}_{\mathrm{II}})Z(\bar{v}_{\mathrm{I}}|\bar{u}_{\mathrm{I}})Z(\mathring{\bar{u}}_{\mathrm{II}}|\mathring{\bar{v}}_{\mathrm{II}})\prod_{s=0}^{n-1}\alpha_s(\bar{v}_{\mathrm{I}}^{(s)})\alpha_s(\mathring{\bar{u}}_{\mathrm{II}}^{(s)}) + reg. \end{aligned} \tag{6.10}$$

Gathering the two contributions, and using the fact that $\bar{u}_{\text{II}} = \mathring{\bar{u}}_{\text{II}}$ in $S_1$ and $\bar{u}_{\text{I}} = \mathring{\bar{u}}_{\text{I}}$ in $S_2$ (and similarly for $\bar{v}$), we get

$$
S(\bar{v}|\bar{u}) = \left(\alpha_p(v_k^{(p)}) - \alpha_p(u_k^{(p)})\right) g(u_k^{(p)}, v_k^{(p)}) \sum_{\text{part}} \Omega(v_k^{(p)}|\mathring{\bar{v}}_{\text{I}}) \Omega(\mathring{\bar{v}}_{\text{II}}|v_k^{(p)}) \Omega(\mathring{\bar{u}}_{\text{I}}|u_k^{(p)}) \Omega(u_k^{(p)}|\mathring{\bar{u}}_{\text{II}})
$$

$$
\times \, \Omega(\mathring{\bar{v}}_{\text{II}}|\mathring{\bar{v}}_{\text{I}}) \Omega(\mathring{\bar{u}}_{\text{I}}|\mathring{\bar{u}}_{\text{II}}) Z(\mathring{\bar{v}}_{\text{I}}|\mathring{\bar{u}}_{\text{I}}) Z(\mathring{\bar{u}}_{\text{II}}|\mathring{\bar{v}}_{\text{II}}) \prod_{s=0}^{n-1} \alpha_s(\mathring{\bar{v}}_{\text{I}}^{(s)}) \alpha_s(\mathring{\bar{u}}_{\text{II}}^{(s)}) + reg.
$$

(6.11)

Expression (6.11) is well-defined in the limit $v_k^{(p)} \to u_k^{(p)}$ and we obtain

$$
\lim_{v_k^{(p)} \to u_k^{(p)}} S(\bar{v}|\bar{u}) = -c\,\alpha_p'(u_k^{(p)}) \sum_{\text{part}} \Omega(u_k^{(p)}|\mathring{\bar{v}}_{\text{I}}) \Omega(\mathring{\bar{v}}_{\text{II}}|u_k^{(p)}) \Omega(\mathring{\bar{u}}_{\text{I}}|u_k^{(p)}) \Omega(u_k^{(p)}|\mathring{\bar{u}}_{\text{II}})
$$

$$
\times \, \Omega(\mathring{\bar{v}}_{\text{II}}|\mathring{\bar{v}}_{\text{I}}) \Omega(\mathring{\bar{u}}_{\text{I}}|\mathring{\bar{u}}_{\text{II}}) Z(\mathring{\bar{v}}_{\text{I}}|\mathring{\bar{u}}_{\text{I}}) Z(\mathring{\bar{u}}_{\text{II}}|\mathring{\bar{v}}_{\text{II}}) \prod_{s=0}^{n-1} \alpha_s(\mathring{\bar{v}}_{\text{I}}^{(s)}) \alpha_s(\mathring{\bar{u}}_{\text{II}}^{(s)}) + reg.,
$$

(6.12)

where $reg$ means the regular terms do not depend on the derivative of $\alpha_p$ when $v_k^{(p)} \to u_k^{(p)}$.

The equation (6.12) can be considered as

$$
\lim_{v_k^{(p)} \to u_k^{(p)}} S(\bar{v}|\bar{u}) = -c\,\alpha_p'(u_k^{(p)})\,\Omega(\mathring{\bar{u}}|u_k^{(p)}) \Omega(\mathring{\bar{v}}|u_k^{(p)})\, S^{(\text{mod})}(\mathring{\bar{v}}|\mathring{\bar{u}}) + reg.
$$

(6.13)

where $S^{(\text{mod})}(\mathring{\bar{v}}|\mathring{\bar{u}})$ is a scalar product of the form (4.11) but with other functions:

$$
\alpha_s^{(\text{mod})}(z) = \frac{\Omega(u_k^{(p)}|z^{(s)})}{\Omega(z^{(s)}|u_k^{(p)})} \alpha_s(z).
$$

(6.14)

More explicitly,

$$
\alpha_s^{(\text{mod})}(z) = \alpha_s(z) \quad \text{for} \quad |s-p| > 1, \qquad \alpha_{p-1}^{(\text{mod})}(z) = \alpha_{p-1}(z)\, f(z, u^{(p)}),
$$

$$
\alpha_p^{(\text{mod})}(z) = \alpha_p(z)\, \frac{\gamma_p(u_k^{(p)}, z)}{\gamma_p(z, u_k^{(p)})}, \qquad \alpha_{p+1}^{(\text{mod})}(z) = \alpha_{p+1}(z)\, \frac{1}{f(u_k^{(p)}, z)}.
$$

(6.15)

Moreover, if we require that $u_k^{(p)}$ satisfies Bethe equation (2.10)

$$
\alpha_p(u_k^{(p)}) \Omega(\mathring{\bar{u}}|u_k^{(p)}) = \Omega(u_k^{(p)}|\mathring{\bar{u}})
$$

(6.16)

we obtain equality

$$
\lim_{v_k^{(p)} \to u_k^{(p)}} S(\bar{v}|\bar{u}) = -c\, \frac{\alpha_p'(u_k^{(p)})}{\alpha_p(u_k^{(p)})}\, \Omega(u_k^{(p)}|\mathring{\bar{u}})\, \Omega(\mathring{\bar{v}}|u_k^{(p)})\, S^{(\text{mod})}(\mathring{\bar{v}}|\mathring{\bar{u}}) + reg.,
$$

(6.17)

where $reg$ means the regular terms does not depend on the derivative of $\alpha_p$ when $v_k^{(p)} \to u_k^{(p)}$.

# 7  Gaudin formula for the norm

The Gaudin formula expresses the norm of on-shell BVs as a determinant of some matrix, the Gaudin matrix. This property has been proven in the context of $Y(\mathfrak{gl}_2)$ models in [13] and in [20] for $Y(\mathfrak{gl}_3)$ models. For $Y(\mathfrak{gl}_n)$ and $Y(\mathfrak{gl}_{m|n})$ models, see [8] for a general presentation. This implies that for $Y(\mathfrak{o}_{2n+1})$ models, the property is proven for BVs of the form $\mathbb{B}(\varnothing, \bar{u}^{(1)}, \ldots, \bar{u}^{(n-1)})$, since they correspond to $Y(\mathfrak{gl}_n)$ BVs. Moreover, since generalized models for $Y(\mathfrak{o}_3)$ are equivalent to those for $Y(\mathfrak{gl}_2)$, see [18], it is also valid for $n = 1$. The goal of this section is to prove that the Gaudin formula is valid for all on-shell BVs in $Y(\mathfrak{o}_{2n+1})$ models.

## 7.1  Gaudin matrix for orthogonal models

We start with the Bethe ansatz equations (BAEs):

$$\alpha_s(\bar{u}_{\mathrm{I}}^{(s)}) \;=\; \frac{f(\bar{u}_{\mathrm{I}}^{(s)}, \bar{u}_{\mathrm{II}}^{(s)})\, f(\bar{u}^{(s+1)}, \bar{u}_{\mathrm{I}}^{(s)})}{f(\bar{u}_{\mathrm{II}}^{(s)}, \bar{u}_{\mathrm{I}}^{(s)})\, f(\bar{u}_{\mathrm{I}}^{(s)}, \bar{u}^{(s-1)})}, \quad s = 1, \ldots, n-1,$$

$$\alpha_0(\bar{u}_{\mathrm{I}}^{(0)}) \;=\; \frac{\mathfrak{f}(\bar{u}_{\mathrm{I}}^{(0)}, \bar{u}_{\mathrm{II}}^{(0)})}{\mathfrak{f}(\bar{u}_{\mathrm{II}}^{(0)}, \bar{u}_{\mathrm{I}}^{(0)})}\, f(\bar{u}^{(1)}, \bar{u}_{\mathrm{I}}^{(0)}),$$

that hold for arbitrary partitions of the sets $\bar{u}^{(s)}$ into subsets $\{\bar{u}_{\mathrm{I}}^{(s)}, \bar{u}_{\mathrm{II}}^{(s)}\}$, with $\bar{u}^{(-1)} = \varnothing = \bar{u}^{(n)}$.

We introduce

$$\Gamma_j^{(s)}(\bar{u}) = \alpha_s(u_j^{(s)})\, \frac{f(\bar{u}_j^{(s)}, u_j^{(s)})\, f(u_j^{(s)}, \bar{u}^{(s-1)})}{f(u_j^{(s)}, \bar{u}_j^{(s)})\, f(\bar{u}^{(s+1)}, u_j^{(s)})}, \quad j = 1, ..., r_s,\; s = 1, ..., n-1,$$

$$\Gamma_j^{(0)}(\bar{u}) = \alpha_0(u_j^{(0)})\, \frac{\mathfrak{f}(\bar{u}_j^{(0)}, u_j^{(0)})}{\mathfrak{f}(u_j^{(0)}, \bar{u}_j^{(0)})\, f(\bar{u}^{(1)}, u_j^{(0)})}, \qquad j = 1, ..., r_0,$$

so that the BAEs for the partitions $\bar{u}^{(s)} \vdash \{u_j^{(s)}, \bar{u}_j^{(s)}\}$ read $\Gamma_j^{(s)}(\bar{u}) = 1$.

The Gaudin matrix $G(\bar{u})$ for the $\mathfrak{o}_{2n+1}$ models is a block matrix $G(\bar{u}) = \big(G^{(s,p)}(\bar{u})\big)_{s,p=0,1,\ldots,n-1}$, where the block $G^{(s,p)}(\bar{u})$ has size $r_s \times r_p$. The entries in each block are defined by

$$G_{j,k}^{(s,p)}(\bar{u}) = -c\,\frac{d}{du_k^{(p)}} \ln \Gamma_j^{(s)}(\bar{u}). \tag{7.1}$$

Let us note that in the context of generalized models the functions $\alpha_s$ are free, so that $\Gamma_j^{(s)}$ is a function depending on two sets of variables $\bar{u}$ and $\bar{\alpha} = \{\alpha_s(u_j^{(s)}), 1 \leq j \leq r_s,\, 0 \leq s \leq n-1\}$, considered as independent variables, so that we should write $\Gamma_j^{(s)}(\bar{u}; \bar{\alpha})$ instead of $\Gamma_j^{(s)}(\bar{u})$. We keep the latter notation to lighten the presentation. In the same way, the matrix $G$ depends on two sets of variables $\bar{u}$ and $\bar{X}$ with

$$X_j^{(s)} = -c\,\frac{d}{dz} \ln \alpha_s(z)\Big|_{z=u_j^{(s)}}, \quad 1 \leq j \leq r_s,\; 0 \leq s \leq n-1.$$

To present the explicit expression of the entries of $G$, we introduce the rational functions

$$K^{(s)}(x,y) = \frac{2c_s c}{(x-y)^2 - c_s^2}, \quad I(x,y) = \frac{c^2}{(x-y+c)(x-y)}, \quad c_s = \Big(1 - \frac{\delta_{s,0}}{2}\Big)c.$$

Then, we have

$$
\begin{aligned}
G_{j,k}^{(s,p)}(\bar{u}) &= 0, \quad \text{when} \quad |s-p| > 1 \,, \\
G_{j,k}^{(s,s-1)}(\bar{u}) &= -\, I(u_j^{(s)}, u_k^{(s-1)}) \,, \\
G_{j,k}^{(s,s+1)}(\bar{u}) &= -\, I(u_k^{(s+1)}, u_j^{(s)}) \,, \\
G_{j,k}^{(s,s)}(\bar{u}) &= \delta_{jk} \left( X_j^{(s)} - \sum_{\ell=1}^{r_s} K^{(s)}(u_j^{(s)}, u_\ell^{(s)}) + \right. \\
&\quad \left. + \sum_{\ell=1}^{r_{s-1}} I(u_j^{(s)}, u_\ell^{(s-1)}) + \sum_{\ell=1}^{r_{s+1}} I(u_\ell^{(s+1)}, u_j^{(s)}) \right) + K^{(s)}(u_j^{(s)}, u_k^{(s)}) \,.
\end{aligned}
\tag{7.2}
$$

Notice that for any $s$ and $j$, we have the relation

$$
\sum_{p=0}^{n-1} \sum_{k=1}^{r_p} G_{j,k}^{(s,p)}(\bar{u}) = X_j^{(s)} \,.
\tag{7.3}
$$

Due to the property of BVs, and the results obtained in the $\mathfrak{gl}_n$ models, we know that $G(\varnothing, \bar{u}^{(1)}, ..., \bar{u}^{(n-1)}) = G_{\mathfrak{gl}_n}(\bar{u}^{(1)}, ..., \bar{u}^{(n-1)})$, where $G_{\mathfrak{gl}_n}(\bar{u})$ is the Gaudin matrix for $\mathfrak{gl}_n$ models, see [8, 13, 19].

## 7.2 Korepin criteria for orthogonal models

The Korepin criteria is a list of property for a series of functions $F^{\bar{r}}$ of $2\|\bar{r}\|$ variables, with $\bar{r} = (r_0, r_1, \ldots, r_{n-1})$ and $\|\bar{r}\| = r_0 + r_1 + \ldots + r_{n-1} > 0$:

**Definition 7.1 (Korepin criteria)** *Let $F^{\bar{r}}$, be a series of functions of $2\|\bar{r}\|$ variables $(\bar{X}; \bar{u})$. We say that the functions $F^{\bar{r}}$ obey the Korepin criteria if they obey the following properties*

1. *The functions $F^{\bar{r}}$ are symmetric under the exchange of any pairs $(X_j^{(s)}, u_j^{(s)}) \leftrightarrow (X_k^{(s)}, u_k^{(s)})$.*

2. *The functions $F^{\bar{r}}$ are polynomials of degree 1 in each $X_j^{(s)}$.*

3. *For $\|\bar{r}\| = 1$ with $s$ the unique color such that $r_s = 1$, we have $F^{\bar{r}}(X_1^{(s)}, u_1^{(s)}) = X_1^{(s)}$.*

4. *The coefficient of $X_j^{(s)}$ in $F^{\bar{r}}(\bar{X}; \bar{u})$ is equal to a function $F^{\bar{r}'}(\bar{Y}; \bar{v})$:*

$$
\frac{\partial F^{\bar{r}}(\bar{X}; \bar{u})}{\partial X_j^{(s)}} = F^{\bar{r}'}(\bar{Y}; \bar{v}) \,,
\tag{7.4}
$$

*where the variables entering the new function $F^{\bar{r}'}(\bar{Y}; \bar{v})$ are expressed as*

$$
\begin{aligned}
\bar{r}' &= (r_0, r_1, ..., r_{s-1}, r_s - 1, r_{s+1}, ..., r_{n-1}), \\
v_k^{(s)} &= u_k^{(s)} \quad ; \quad Y_k^{(s)} = X_k^{(s)} - K^{(s)}(u_j^{(s)}, u_k^{(s)}), \quad 1 \le k < j \,, \\
v_k^{(s)} &= u_{k+1}^{(s)} \quad ; \quad Y_k^{(s)} = X_{k+1}^{(s)} - K^{(s)}(u_j^{(s)}, u_{k+1}^{(s)}), \quad j \le k < r_s \,, \\
v_k^{(s+1)} &= u_k^{(s+1)} \quad ; \quad Y_k^{(s+1)} = X_k^{(s+1)} + I(u_k^{(s+1)}, u_j^{(s)}), \quad 1 \le k \le r_{s+1} \,, \\
v_k^{(s-1)} &= u_k^{(s-1)} \quad ; \quad Y_k^{(s-1)} = X_k^{(s-1)} + I(u_j^{(s)}, u_k^{(s-1)}), \quad 1 \le k \le r_{s-1} \,, \\
v_k^{(p)} &= u_k^{(p)} \quad ; \quad Y_k^{(p)} = X_k^{(p)}, \quad 1 \le k \le r_p, \quad |s-p| > 1 \,.
\end{aligned}
\tag{7.5}
$$

5. $F^{\bar{r}}(\bar{X}; \bar{u}) = 0$ if all $X_j^{(s)} = 0$.

The Korepin criteria allows a characterization of the functions $F^{\bar{r}}$:

**Lemma 7.2** *The Korepin criteria fixes uniquely the functions $F^{\bar{r}}$.*

*Proof:* Let $F^{\bar{r}}$ and $\tilde{F}^{\bar{r}}$ be two series of functions obeying the Korepin criteria. We prove by recursion on $\|\bar{r}\|$ that the functions $F^{\bar{r}}$ and $\tilde{F}^{\bar{r}}$ coincide.

By point 3. we have immediately that for $\|\bar{r}\| = 1$, $F^{\bar{r}} = \tilde{F}^{\bar{r}}$. Suppose now that when $\|\bar{r}\| \le r$, we have $F^{\bar{r}} = \tilde{F}^{\bar{r}}$ and consider $\bar{r}$ with $\|\bar{r}\| = r + 1$, $F^{\bar{r}}$ and $\tilde{F}^{\bar{r}}$.

By point 4. and the recursion hypothesis, we deduce that

$$\frac{\partial}{\partial X_j^{(s)}} \left( F^{\bar{r}}(\bar{X}; \bar{u}) - \tilde{F}^{\bar{r}}(\bar{X}; \bar{u}) \right) = 0 \,,$$

Since by point 2. the functions are of degree 1 in $X_j^{(s)}$, it implies that they coincide up to a term independent from $\bar{X}$. Finally, point 5. shows that they coincide exactly at $\bar{X} = 0$, which ends the proof. $\qquad\qquad\square$

For our purpose, we define the functions

$$F^{\bar{r}}(\bar{X}; \bar{u}) = \det G(\bar{u})\,, \quad \|\bar{r}\| = |\bar{u}| = |\bar{X}|\,. \tag{7.6}$$

For generalized models, where no specific representations have been chosen, the variables $\bar{X}$ are independent from the variables $\bar{u}$, so that $F^{\bar{r}}$ is a true function of $2\|\bar{r}\|$ variables.

**Proposition 7.3** *The functions $F^{\bar{r}}$ defined in (7.6) satisfy the Korepin criteria.*

*Proof:* We prove the properties of the criteria point by point.

Point 1: The exchange of any pair amounts to exchange the two corresponding lines and the two corresponding column in the matrix $G$. Then, since each $F^{\bar{r}}$ is a determinant, it is invariant under this exchange.

Points 2. and 3. are obvious from the explicit form (7.2) of $G$.

To get the point 4, when computing $\frac{\partial F^{\bar{r}}(\bar{X}; \bar{u})}{\partial X_j^{(s)}}$, we expand the determinant along the column containing $X_j^{(s)}$, showing that $\frac{\partial F^{\bar{r}}(\bar{X}; \bar{u})}{\partial X_j^{(s)}}$ is just the minor of the diagonal element containing $X_j^{(s)}$. It is thus a determinant of the same form, but with a modification of the variables $X_k^{(p)}$ and $u_k^{(p)}$ as mentioned. The shift of the variables $Y_k^{(p)}$ allows to add the terms $K^{(s)}(u_j^{(s)}, u_k^{(s)})$, $I(u_k^{(s+1)}, u_j^{(s)})$ and $I(u_j^{(s)}, u_k^{(s-1)})$ that should appear in the sums occurring in the diagonal terms.

Point 5. follows from the relation (7.3), which implies the vanishing of $\det G$. $\qquad\square$

## 7.3 Norm of on-shell BVs for orthogonal models

**Theorem 7.4** *Let $\mathbb{B}(\bar{u})$ be an on-shell Bethe vector, and $\mathbb{C}(\bar{v})$ a dual Bethe vector with $|\bar{v}| = |\bar{u}|$. Then,*

$$
\begin{aligned}
\lim_{\bar{v} \to \bar{u}} S(\bar{v}|\bar{u}) &= \lim_{\bar{v} \to \bar{u}} \mathbb{C}(\bar{v})\mathbb{B}(\bar{u}) = \prod_{s,p=0}^{n-1} \prod_{k=1}^{r_s} \prod_{l=1}^{r_p} \Omega(u_k^{(s)}|u_l^{(p)}) \det G(\bar{u}) \\
&= \prod_{s=0}^{n-1} \prod_{\substack{k,l=1 \\ k \ne l}}^{r_s} \gamma_s(u_k^{(s)}, u_l^{(s)}) \prod_{s=1}^{n-1} \frac{h(\bar{u}^{(s)}, \bar{u}^{(s-1)})}{g(\bar{u}^{(s)}, \bar{u}^{(s-1)})} \det G(\bar{u})\,,
\end{aligned}
\tag{7.7}
$$

*where $G(\bar{u})$ is the Gaudin matrix (7.1).*

*Proof:*

Let

$$\mathcal{N}(\bar{u}) = \prod_{s,p=0}^{n-1} \prod_{k=1}^{r_s} \prod_{l=1}^{r_p} \Omega(u_k^{(s)}|u_l^{(p)})^{-1} \lim_{\bar{v}\to\bar{u}} \mathbb{C}(\bar{v})\mathbb{B}(\bar{u}), \tag{7.8}$$

where $\bar{u}$ satisfy Bethe equation (2.10).

From lemma 7.2 and property 7.3, it is sufficient to prove that the $\mathcal{N}(\bar{u})$ obeys the Korepin criteria in the limit $\bar{v}\to\bar{u}$.

- Point 1. It is a direct consequence of the symmetry property for the Bethe vectors.

- Point 2. It is the result of the equation (6.17).

- Point 3. We start with the Bethe vectors and dual Bethe vectors

$$\mathbb{B}(u_1^{(s)}) = \frac{T_{s,s+1}(u_1^{(s)})}{\lambda_{s+1}(u_1^{(s)})}|0\rangle \quad \text{and} \quad \mathbb{C}(v_1^{(s)}) = \langle 0|\frac{T_{s+1,s}(v_1^{(s)})}{\lambda_{s+1}(v_1^{(s)})},$$

  where for $\bar{u} = \{\varnothing,\dots,\varnothing,u_1^{(s)},\varnothing,\dots,\varnothing\}$, we wrote $\mathbb{B}(u_1^{(s)})$ for $\mathbb{B}(\bar{u})$ and similarly for $\mathbb{C}(v_1^{(s)})$. From the commutation relations (2.4) and the conditions (2.7), it is easy to get

$$\mathbb{C}(v_1^{(s)})\mathbb{B}(u_1^{(s)}) = -c\,\frac{\alpha_s(u_1^{(s)}) - \alpha_s(v_1^{(s)})}{u_1^{(s)} - v_1^{(s)}}$$

  which leads to

$$\lim_{v_1^{(s)}\to u_1^{(s)}} \mathbb{C}(v_1^{(s)})\mathbb{B}(u_1^{(s)}) = -c\,\frac{d}{dz}\alpha_s(z)\Big|_{z=u_1^{(s)}} = \alpha_s(u_1^{(s)})X_1^{(s)}.$$

  Now, the Bethe equations (2.10) in this case read $\alpha_s(u_1^{(s)}) = 1$, so that the scalar product is exactly $X_1^{(s)}$ in the limit $v_1^{(s)}\to u_1^{(s)}$. Moreover, since there is only one Bethe parameter, the normalisation factor in (7.8) is 1, so that we get the result.

- Point 4. It follows directly from formula (6.17), taking into account the normalization in (7.8).

- Point 5. When $X_j^{(s)} = 0$, $\alpha_s(v_j^{(s)}) = \alpha_s(u_j^{(s)}) + O\big((v_j^{(s)} - u_j^{(s)})^2\big)$, we can replace $\alpha_s(v_j^{(s)})$ by $\alpha_s(u_j^{(s)})$, so that the sum formula (4.11) rewrites

$$S(\bar{v}|\bar{u}) = \left(\prod_{s=0}^{n-1} \alpha_s(\bar{u}^{(s)})\right) \lim_{\bar{v}\to\bar{u}} S_0(\bar{v}|\bar{u}),$$

$$S_0(\bar{v}|\bar{u}) = \sum_{\text{part}} Z(\bar{v}_{\mathrm{I}}|\bar{u}_{\mathrm{I}})\, Z(\bar{u}_{\mathrm{II}}|\bar{v}_{\mathrm{II}}) \tag{7.9}$$

$$\times \prod_{s=0}^{n-1} \gamma_s(\bar{u}_{\mathrm{I}}^{(s)}, \bar{u}_{\mathrm{II}}^{(s)})\, \gamma_s(\bar{v}_{\mathrm{II}}^{(s)}, \bar{v}_{\mathrm{I}}^{(s)}) \prod_{s=0}^{n-1} \frac{h(\bar{u}_{\mathrm{II}}^{(s+1)}, \bar{u}_{\mathrm{I}}^{(s)})h(\bar{v}_{\mathrm{I}}^{(s+1)}, \bar{v}_{\mathrm{II}}^{(s)})}{g(\bar{u}_{\mathrm{I}}^{(s+1)}, \bar{u}_{\mathrm{II}}^{(s)})g(\bar{v}_{\mathrm{II}}^{(s+1)}, \bar{v}_{\mathrm{I}}^{(s)})}.$$

  $S_0(\bar{v}|\bar{u})$ is a particular case of scalar product where all $\alpha_s(z) = 1$. This corresponds to a particular model where $T_{i,j}(z) = \delta_{ij}$. But in that case, the only non zero Bethe vector is $|0\rangle$, which is not considered here[5]. This implies that $S_0(\bar{v}|\bar{u}) = 0$ as soon as $|\bar{u}| > 0$, and proves Point 5. $\qquad\square$

---

[5]We recall that the Korepin criteria applies to functions $F^{\bar{r}}$ with $\|\bar{r}\| > 0$.

# A  Proof of the lemma 4.2

We prove this lemma using a recursion on $n$, the rank of $\mathfrak{o}_{2n+1}$. Since the property has been already proved for $\mathfrak{gl}_2$, it is also true for $\mathfrak{o}_3$, i.e. when $n = 1$.

Through iterative applications of the relation (3.19) with $\ell = n - 1$, we can express a dual BV as

$$
\mathbb{C}(\bar{v}^{(0)}, \ldots, \bar{v}^{(n-1)}) = \sum_{j_1, \ldots, j_{r_{n-1}}} \sum_{\text{part}} \Psi^{(n-1)}_{j_1 \ldots j_{r_{n-1}}}(\bar{v}^{(0)}, \ldots, \bar{v}^{(n-1)})
$$

$$
\times \, \mathbb{C}(\bar{v}^{(0)}_{\mathrm{I\!I}}, \ldots, \bar{v}^{(n-2)}_{\mathrm{I\!I}}) \, \frac{T_{n,j_{r_n}}(v^{(n-1)}_{r_{n-1}}) \cdots T_{n,j_1}(v^{(n-1)}_1)}{\lambda_n(\bar{v}^{(n-1)})} , \tag{A.1}
$$

where $\Psi^{(n-1)}_{j_1 \ldots j_{r_{n-1}}}(\bar{v}^{(0)}, \ldots, \bar{v}^{(n-1)})$ are rational functions that do not depend on the eigenvalues $\lambda_j(v)$, $j = -n, \ldots, n$, where $r_{n-1} = \left| \bar{v}^{(n-1)} \right|$. In (A.1), the sum runs on all values of $j_1, \ldots, j_{r_{n-1}} = -n, \ldots, n - 1$ and over partitions of the sets $\bar{v}^{(s)} \vdash \{\bar{v}^{(s)}_{\mathrm{I}}, \bar{v}^{(s)}_{\mathrm{I\!I}}\}$, $s = 0, \ldots, n - 1$.

In the same way, from the action formula (3.6), we get by iteration

$$
\frac{T_{n,j_{r_{n-1}}}(v^{(n-1)}_{r_{n-1}}) \cdots T_{n,j_1}(v^{(n-1)}_1)}{\lambda_n(\bar{v}^{(n-1)})} \mathbb{B}(\bar{u}) = \sum_{\text{part}} \left( \prod_{s=0}^{n-1} \alpha_s(\bar{w}^{(s)}_{\mathrm{I\!I\!I}}) \right) \Phi_{j_1, \ldots, j_{r_{n-1}}}(\bar{w}) \, \mathbb{B}(\bar{w}_{\mathrm{I\!I}}) , \tag{A.2}
$$

with now $\bar{w}^{(s)} = \{\bar{u}^{(s)}, \bar{v}^{(n-1)}, \bar{v}^{(n-1)} - c(s - 1/2)\}$ with $s = 0, \ldots, n - 2$. This shows that the functions $\alpha_s$ appearing in the r.h.s. of (A.2) do not depend on the parameters $\{\bar{v}^{(0)}, \ldots, \bar{v}^{(n-2)}\}$.

Moreover, each operator $T_{n,j}(u)$ contributes to the color $n - 1$ by a factor

$$
\begin{cases} -2, & \text{if } j = -n, \\ -1, & \text{if } |j| < n. \end{cases}
$$

Hence, the r.h.s. in (A.2) is formed of BVs with no color $n - 1$, and as such corresponds to an $\mathfrak{o}_{2n-1}$ model with Bethe parameters $\bar{w}_{\mathrm{I\!I}} \subset \{\bar{u}^{(s)}, \bar{v}^{(n-1)}, \bar{v}^{(n-1)} - c(s - 1/2)\}$. Then, by the recursion hypothesis, we know that the scalar products $\mathbb{C}(\bar{v}^{(0)}_{\mathrm{I\!I}}, \ldots, \bar{v}^{(n-2)}_{\mathrm{I\!I}}) \mathbb{B}(\bar{w}_{\mathrm{I\!I}})$ depend only on $\alpha_s(\bar{v}^{(s)}_{\mathrm{I\!I}})$ and $\alpha_s(\bar{w}^{(s)}_{\mathrm{I\!I}})$. This shows that for $s = 0, \ldots, n - 2$, the set $\bar{v}^{(s)}$ enter only in the functions $\alpha_s$. This property is preserved by the product occurring in the r.h.s. of (A.2). By symmetry in the exchange $\bar{u} \leftrightarrow \bar{v}$ this property also applies to the sets $\bar{u}^{(s)}$, $s = 0, \ldots, n - 2$.

It remains to show that the property is also valid for the sets $\bar{u}^{(n-1)}$ and $\bar{v}^{(n-1)}$. For such a purpose, we perform the same calculation as above with the recursion (3.19) for an integer $\ell$ such that[6] $\ell \neq n - 1$ and $|\bar{v}^{(\ell)}| \neq 0$. To simplify the presentation, we take $\ell = n - 2$, the other cases being similar.

$$
\mathbb{C}(\bar{v}^{(0)}, \ldots, \bar{v}^{(n-2)}, \bar{v}^{(n-1)}) =
$$

$$
\sum_{\substack{-n \leq j_1, \ldots, j_{r_{n-2}} \leq n-2 \\ n-1 \leq i_1, \ldots, i_{r_{n-2}} \leq n}} \sum_{\text{part}} \alpha_{n-1}(\bar{v}^{(n-1)}_{\mathrm{I\!I\!I}}) \Psi^{(n-2)}_{\bar{i}, \bar{j}}(\bar{v}) \, \mathbb{C}(\bar{v}^{(0)}_{\mathrm{I\!I}}, \ldots, \bar{v}^{(n-3)}_{\mathrm{I\!I}}, \varnothing, \bar{v}^{(n-1)}_{\mathrm{I\!I}}) \mathcal{O}_{\bar{i}, \bar{j}}(\bar{v}^{(n-2)}) ,
$$

$$
\mathcal{O}_{\bar{i}, \bar{j}}(\bar{v}^{(n-2)}) = \frac{T_{i_1, j_1}(v^{(n-2)}_1) \cdots T_{i_{r_{n-2}}, j_{r_{n-2}}}(v^{(n-2)}_{r_{n-2}})}{\lambda_{n-1}(\bar{v}^{(n-2)})} ,
$$

$$
\tag{A.3}
$$

---

[6]Note that if such $\ell$ does not exist, we are dealing with $\mathfrak{gl}_2$ BVs with Bethe parameters $\bar{v}^{(n-1)}$, for which the lemma 4.2 has been already proven.

where $\bar{\imath}$ (resp. $\bar{\jmath}$) stands for $i_1, ..., i_{r_{n-2}}$ (resp. $j_1, ..., j_{r_{n-2}}$).

It remains to act with $\mathcal{O}_{\bar{\imath},\bar{\jmath}}(\bar{v}^{(n-2)})$ on $\mathbb{B}(\bar{u})$ using iteratively the action formula (3.6). As fas as $\alpha_s$ functions are concerned, it produces terms $\prod_{s=0}^{n-1} \alpha_s(\bar{w}_{\mathrm{I}}^{(s)})$, where $\bar{w}^{(s)} = \{\bar{u}^{(s)}, \bar{v}^{(n-2)}, \bar{v}^{(n-2)} - c(s - 1/2)\}$. This shows that the set $\bar{v}^{(n-1)}$ enters only in the $\alpha_{n-1}$ function, as expected. By symmetry in the exchange $\bar{u} \leftrightarrow \bar{v}$ this property also true for the set $\bar{u}^{(n-1)}$.

Finally, from the construction done above, it is clear that each Bethe parameter $v_i^{(s)}$ and each Bethe parameter $u_j^{(s)}$ occurs at most once in $\alpha^{(s)}$. $\qquad\square$

# B  Existence of a suitable model

We first remind that from Theorem 5.16 in [1], the finite dimensional irreducible modules of $Y(\mathfrak{o}_{2n+1})$ are classified by monic polynomials $P_0(u)$, ..., $P_{n-1}(u)$ such that

$$\frac{\lambda_s(u)}{\lambda_{s+1}(u)} = \frac{P_s(u+c)}{P_s(u)}, \ \ s = 1, \ldots, n-1 \ \ \text{and} \ \ \frac{\lambda_0(u)}{\lambda_1(u)} = \frac{P_0(u+c/2)}{P_0(u)}. \tag{B.1}$$

We are looking for a module where relations (4.5) are obeyed. We denote $p_s = |\bar{u}_{\mathrm{i}}^{(s)}|$ and $q_s = |\bar{v}_{\mathrm{ii}}^{(s)}|$. Let us consider two irreducible modules:

1. One with polynomials

$$P_s(z) = \prod_{k=1}^{p_s} (z - c - u_k^{(s)}), \ \ s = 1, \ldots, n \ \ \text{and} \ \ P_0(z) = \prod_{k=1}^{p_0} (z - c/2 - u_k^{(0)}), \tag{B.2}$$

   where $\bar{u}_{\mathrm{i}}^{(s)} = \{u_k^{(s)}, \ k = 1, \ldots, p_s\}$, and associated to the monodromy matrix $T^{(1)}(u)$.

2. One with polynomials

$$Q_s(z) = \prod_{k=1}^{q_s} (z - c - v_k^{(s)}), \ \ s = 1, \ldots, n \ \ \text{and} \ \ Q_0(z) = \prod_{k=1}^{q_0} (z - c/2 - v_k^{(0)}), \tag{B.3}$$

   where $\bar{v}_{\mathrm{ii}}^{(s)} = \{v_k^{(s)}, \ k = 1, \ldots, q_s\}$, and associated to the monodromy matrix $T^{(2)}(u)$.

We consider the composite model based on $T^{(1)}(u)$ and $T^{(2)}(u)$. By construction, we have

$$\begin{aligned} \alpha_s^{[1]}(u) &= \frac{Q_s(u+c)}{Q_s(u)}, \ \ s = 1, \ldots, n-1 \ \ \text{and} \ \ \alpha_0^{[1]}(u) = \frac{Q_0(u+c/2)}{Q_0(u)}, \\ \alpha_s^{[2]}(u) &= \frac{P_s(u+c)}{P_s(u)}, \ \ s = 1, \ldots, n-1 \ \ \text{and} \ \ \alpha_0^{[2]}(u) = \frac{P_0(u+c/2)}{P_0(u)}. \end{aligned} \tag{B.4}$$

In view of the expressions (B.2) and (B.3), for generic values of the Bethe parameters, we clearly have

$$\alpha_s^{[1]}(z) = 0 \ \ \text{if} \ \ z \in \bar{v}_{\mathrm{ii}}^{(s)} \ \ \text{and} \ \ \alpha_s^{[2]}(z) = 0 \ \ \text{if} \ \ z \in \bar{u}_{\mathrm{i}}^{(s)}, \tag{B.5}$$

as required.

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
