# Peer review of "Scalar products and norm of Bethe vectors in $\mathfrak{o}_{2n+1}$ invariant integrable models"

_SciPost Physics_

## Round 1 · Referee Report · Anonymous (Referee 1) · 2025-4-18

Strengths

  1. The paper gives a complete solution of a complicated and essential problem.
  2. Detailed presentation of a complicated proof.
  3. Explanation of the relations with the known cases, namely the $Y(\mathfrak{0}_3)$ case.
  4. Extremely efficient choice of notations making several steps of the proof readable

Weaknesses

  1. Lack of presentation of the context and possible future developments

Report

It is a very interesting and important paper for every reader following the previous work of the authors. Generalisation of the results for norms of the Bethe vectors to new algebras and a complete proof of the Gaudin formula in new cases is highly non-trivial and absolutely necessary program for the field of quantum integrability. The paper is very technical but it is the feature of the problem. I appreciated the detailed presentation of the proof, good presentation of the result (including the explicit form of the Gaudin matrix) and separate section on the simplest case $Y(\mathfrak{0}_3)$ where previously known results are recovered.

Requested changes

  1. I would appreciate a more detailed introduction with more context and explanation of the importance of the problem as well as a conclusion indicating future possible developments (scalar products of off-shell on-shell Bethe vectors? generalisation of the Gaudin formula to other series?)

Recommendation

Publish (easily meets expectations and criteria for this Journal; among top 50%)

---

## Round 1 · Referee Report · Anonymous (Referee 2) · 2025-4-28

Strengths

1) Present a breakthrough on a previously-identified and long-standing research stumbling block. 2) The results derived are original and easily formulated and reproduced $gl_n$ known results, when opportunely specialized, of particular interest is the block determinant formula for the norm.

Weaknesses

There are no conclusions and the intro and references are too briefly addressed.

Report

Dear Editor,
This is an important paper where the authors continue the relevant analysis of higher rank quantum integrable models. They generalize results that they have previously obtained for the models associated to the $gl_{n}$ and $gl_{n,m}$ symmetry in collaboration with fellow researchers.
Here, they derive scalar products of the off-shell Bethe vectors as sum of partitions over Bethe parameters for the class of quantum integrable models associated to $o_{2n+1}$ higher rank symmetry. They show that the building blocks of these scalar products are the so-called highest coefficients, for which they prove recurrence relations and residue theorems that completely characterize them and that they verify reproduce the known $gl_{n}$ results, once the reduction to these last models is implemented. Notably, they are able to prove a block determinant form for the Gaudin norm of on-shell Bethe vectors of the $o_{2n+1}$ model, in this way extending results known by the same research group only for n=1.
The research material is made accessible to the general public with clear notations and explanations, this make possible to follow their proofs and argue in favor of their results even if these are somewhere quite involved.
The authors have indicated for the fulfilments of journal expectations, a breakthrough on a previously-identified and long-standing research problems, I agree on it and I think that it deserves to be published once the authors will take into accounts the following clarification requests:
a) In section 5.2, before Proposition 5.4, the authors implement a specialization on the set of the partitions of the variables putting the sets of the u and v with label 0 to the empty sets. While it is possible, following their arguments, to understand how the recurrences adapt to this specialization and rederive those of the $gl_{n}$ case, what for me is not well described is why is this specialization that bring from the $o_{2n+1}$ models to the $gl_{n}$ models. Similarly, in the intro to section 7, in line 5, it is written that “BVs of the form …, since they correspond to $Y(gl_{n})$ BVs”. I think that it would be important to comment why these BVs belongs to $Y(gl_{n})$ and why the above specification allows to go from models associated to $o_{2n+1}$ to models associated to $gl_{n}$. Indeed, this is central to understand why the authors can reproduce from the $o_{2n+1}$ case the recursion relations and block determinant of the Gaudin norm for the $gl_{n}$ case.
b) The paper misses to define the main prospectives and research directions that can emerge as consequences of the results here derived. I think that the authors should write at least a conclusion. Even the current introduction is very short in defining the research background where this research is located and the use of this type of results, e.g. when the authors refer to correlation functions, they only cite the book in citation [3], where only the primordial results on these quantities are described, while the main present and fundamental results in the Algebraic Bethe Ansatz using the Slavnov’s scalar product formula [22] are missing.
c) In the integrable models associated to $gl_3$ symmetry, there are block determinant formulae for the scalar products of two Bethe vectors associated two a periodic and twisted boundary condition, respectively. These formulae are fundamental as they are extremely easier of the sum over partition ones and they have been used moreover to generate results on dynamical objects like the form factors, all these are results generated by some/all the present authors with fellow researchers. It is then natural to ask if there is the possibility to have some block determinant formula also for the scalar products of models associated to $o_{2n+1}$, for n=2 or 3 in view of the specialization to the corresponding $gl_n$ models. If this is the case, then, the next natural question is if the authors have planned or can already state the possibility to study form factors of local operators. In the current paper is only discussed the $o_3$ case due to an isomorphism to $gl_2$.
d) It seems to me that the functions $\lambda_{i}$ are not arbitrary free functions, as they have to satisfy the condition (2.8); the “formal parameter” in (3.7) seems to be z and not u as stated before this equation.

Requested changes

The requested changes are listed in the report.

Recommendation

Publish (meets expectations and criteria for this Journal)

---

## Round 1 · Referee Report · Anonymous (Referee 3) · 2025-5-2

Strengths

The result for the "norm" is explicit.

Weaknesses

1) The paper is very technical. 2) The basic definitions at the beginning of the paper seem incomplete or need more explanation, at least for readers who have not worked on the subject themselves. 3) Little effort is made to explain the physical and historical context of the results.

Report

The authors consider Bethe vectors connected with the Yangian $Y(\mathfrak{o}(2n + 1))$ defined by eqs. (2.1), (2.2) of their manuscript. These are vectors connected with a class of representations characterized by (2.7), (2.8). The issue of the existence of such representations is considered in Appendix B. The authors introduce a concept of duality on $Y(\mathfrak{o}(2n + 1))$ in eqn. (2.12) and extend it to the considered representations. They define `scalar products' as pairings of Bethe states and their duals. This is quite common in this field and probably goes back to the seminal paper [13].

For the scalar products they find a formula representing them as sums over partitions of the two sets of Bethe rapidities that parameterize the Bethe vectors, eqn. (4.11), Theorem 4.6. This is the first main result of the paper. The structure of the formula is the same as in the $Y(\mathfrak{gl}_n)$ case [13,20,8] which is interesting. One should, however, say that the formula is not quite explicit, as the so-called highest coefficients appearing in the formula are not explicitly given, but rather characterized by certain properties. The second main results is formulated in Theorem 7.4. It says that if the two rapidity sets that parameterize the Bethe vectors are equal and satisfy the Bethe equations, then the scalar product is proportional (with an explicit factor) to the Gaudin matrix associated with the Bethe equations. Such a connection is expected to hold in general. Here it is proved along the lines of [13] for the models connected with $Y(\mathfrak{o}(2n + 1))$.

Both, theorems 4.6 and 7.4, were, to my best knowledge, unknown before and may be interesting for experts. For this reason I recommend the manuscript for publication.

I suggest, however, that the authors try to improve the readability of the manuscript by clarifying certain points listed below and by adding a summary or conclusion section.

Requested changes

As it stands the introduction reads as if the authors do not care much about the use of their results. They write, for instance, "These vectors are foundational for understanding the spectrum of integrable models and play a central role in calculating physical observables such as correlation functions [3]." But the spectrum is determined by the eigenvalues of the transfer matrix rather than by the eigenvectors. Moreover, many techniques have been developed by people like R. Baxter that make it possible to obtain the spectrum without constructing the eigenvectors (e.g. the construction of the Q-operator and TQ equations). Bethe vectors may be used for two purposes. First, for the calculation of matrix elements of local operators for finite-size or finite temperature lattice models. And second, within the axiomatic form factor approach, for an efficient description of form factors of integrable quantum field theories. It is the first purpose for which the authors work may become useful as a pre-study. If they agree, they should state this more carefully. The authors continue "The scalar product of Bethe vectors is particularly important, as it provides overlaps between eigenstates, allowing the evaluation of matrix elements of operators." Well, the overlaps between eigenstates are either zero or equal to the norm calculated by the authors. Such overlaps do not carry much interesting information. More interesting would be overlaps between eigenstates and non-eigenstates of Bethe form. For such overlaps there exists a nice determinant formula [22] which is the reason why correlation functions of $U_q (\mathfrak{sl}_2)$ related models could be studied, to a certain extend, by Bethe ansatz methods. For the case at hand it seems to me that we are still far away from similarly efficient formulae. Perhaps the authors could say something about it. I am sure that many readers would appreciate it. Their next sentence is "Such calculations are crucial for studying the dynamics and thermodynamics of integrable systems [22]." I disagree again. The partition function and hence the thermodynamics of integrable and non-integrable models depends only on the spectrum. It can be obtained from the thermodynamic Bethe ansatz or from the quantum transfer matrix formalism without any recourse to Bethe vectors or overlaps. Overlaps (in determinantal form) are indeed useful for calculating correlation functions as can be learned from [22]. This is true for the static and for the dynamical case, in the ground state, for the system coupled to a heat bath and even in non-equlibrium settings.

I could continue this way, sentence by sentence. In my opinion the introduction is not well written and not to the point. Also no word is said, why we should care about the $\mathfrak{o} (n)$ symmetric models. As far as I can judge about it, $\mathfrak{o} (n)$ spin chains are of little interest in physical applications. $\mathfrak{o} (n)$ symmetric models rather appear in the context of integrable quantum field theories, where $\mathfrak{o} (n)$ sigma models and $\mathfrak{o} (n)$ invariant Gross-Neveu models have attracted attention. In my understanding this is where the interest in $\mathfrak{o} (n)$ symmetric integrable models comes from. Please see e.g. the work "Bethe Ansatz and exact form factors of the $O(N)$ Gross Neveu-model", JHEP 02 (2016) 042 by H. Babujian et al. and the literature cited therein. You will find there, among other interesting references, the reference to the original algebraic Bethe ansatz solution of the $O(n)$ models for even $n$ by de Vega and Karowski in 1987.

My suggestion is to make an effort to improve the introduction for the readers sake and to add a summary, in which it is stated which are the main results of the submitted paper and what might be their use.

Now for some more technical points. I suggest to explain how the transposition defined in (2.3) acts on the monodromy matrix. The authors do provide a similar explanation for the transposition (2.11) further below. Actually, that one and also the first proof on top of page 9 appears strange to me. It seems to me that the condition $\langle 0 | = |0\rangle'$ is a restriction of the possible representations. Perhaps the authors can comment on that. Below (2.7): "Arbitrary free functions" do not necessarily satisfy (2.8). The word free appears again further down on the same page. In the equation on the bottom of page 2 there should probably be a bar over $u_1$ and $u_2$. On top of the same page: "On takes a tensor of $[\dots]$" should be "On takes a tensor product of $[\dots]$". On page 3 the off-shell Bethe vectors are defined below (2.10). Do I understand correctly that the Bethe vectors are not just any polynomials in the elements in the upper triangular part of the monodromy matrix acting on the reference state? If yes, please give a full and precise description of your off-shell Bethe vectors. Otherwise add some explanation. In (3.2) $\Omega$ is defined for partitions into pairs of subsets, but in (3.3) you partition into triples. Above (3.7) "the formal parameter $u$". There is no $u$ in eq. (3.7). A reference to Michel Gaudin for the Gaudin matrix would appear fair to me.

Recommendation

Ask for minor revision

---

## Editorial Decision

resubmitted